# (Implicit) Ensembles of Ensembles: Epistemic Uncertainty Collapse in Large Models

Andreas Kirsch                                                                    *blackhc@gmail.com*

**Reviewed on OpenReview:** *https://openreview.net/forum?id=ON7dtdEHVQ*

## Abstract

Epistemic uncertainty is crucial for safety-critical applications and data acquisition tasks. Yet, we find an important phenomenon in deep learning models: an *epistemic uncertainty collapse* as model complexity increases, challenging the assumption that larger models invariably offer better uncertainty quantification. We introduce *implicit ensembling* as a possible explanation for this phenomenon. To investigate this hypothesis, we provide theoretical analysis and experiments that demonstrate uncertainty collapse in explicit *ensembles of ensembles* and show experimental evidence of similar collapse in wider models across various architectures, from simple MLPs to state-of-the-art vision models including ResNets and Vision Transformers. We further develop *implicit ensemble extraction* techniques to decompose larger models into diverse sub-models, showing we can thus recover epistemic uncertainty. We explore the implications of these findings for uncertainty estimation.

## 1 Introduction

Bayesian deep learning provides us with a principled framework for quantifying uncertainty in complex machine learning models (MacKay, 1992; Neal, 1994). A wide range of applications relies on accurate uncertainty estimates. These include active learning (Settles, 2009; Osband et al., 2022), where uncertainty guides data acquisition; anomaly detection (Tagasovska & Lopez-Paz, 2019; Mukhoti et al., 2023), where uncertainty can signal out-of-distribution inputs; and safety-critical systems, where understanding a model's confidence is crucial for responsible deployment (Band et al., 2021).

A key concept in this framework is *epistemic uncertainty*, which represents a model's uncertainty about its predictions due to limited knowledge or data (Smith & Gal, 2018; Der Kiureghian & Ditlevsen, 2009). This form of uncertainty is distinct from *aleatoric uncertainty*, which captures inherent noise or randomness in the data (Kendall & Gal, 2017). Epistemic uncertainty is usually quantified using the mutual information between the model's predictions and its parameters (Houlsby et al., 2011; Gal et al., 2017). For deep ensembles (Lakshminarayanan et al., 2017), which consist of multiple independently trained neural networks, this amounts to the mutual information between the ensemble member index and the predictions. This is equivalent to the disagreement among ensemble members on a particular input (Smith & Gal, 2018): if the models in the ensemble produce widely varying predictions, this is indicative of high epistemic uncertainty.

As deep learning models grow in size and complexity, the quality of their uncertainty estimates generally improves (Tran et al., 2022). However, our work provides evidence for a simple yet important phenomenon: when constructing higher-order ensembles, *ensembles of ensembles*, we observe an *epistemic uncertainty collapse*[1]. This collapse occurs because individual ensembles, given sufficient size and training, converge to similar predictive distributions, causing inter-ensemble disagreement to vanish as the ensemble size grows.

---

[1]We initially observed this behavior in 2021, published informally at https://blackhc.notion.site/Ensemble-of-Ensembles-Epistemic-Uncertainty-for-OoD-4a8df73b0d8942c8872aab1848a4393b, and have since confirmed it in multiple independent experiments that we share here.

Figure 1: **Epistemic Uncertainty Collapse in a Toy Regression Problem.** As the sub-ensemble size increases, epistemic uncertainty vanishes. Ensembles of 10 sub-ensembles with different sub-ensemble sizes. *Left:* True function, data, and ensemble predictions. *Middle:* Epistemic uncertainty across input space. *Right:* Mean epistemic uncertainty vs. sub-ensemble size.

We hypothesize that, similar to ensembles of ensembles, *implicit ensembling* might occur within large neural networks, potentially leading to significant underestimation of epistemic uncertainty for traditional uncertainty estimators that rely on the final logits. Hence, similar to deep ensembles that have been found to offer better calibration (Ovadia et al., 2019), implicit ensembling may explain why larger models also appear more calibrated (Tran et al., 2022). A recent preprint by Fellaji & Pennerath (2024) provides additional evidence of this phenomenon occurring even in simple over-parameterized MLPs trained on standard benchmark datasets but does not offer an explanation.

We introduce implicit ensembling as a possible explanation for this phenomenon. Our theoretical analysis demonstrates uncertainty collapse in ensembles of ensembles, and we present experimental evidence of similar collapse in wider models. Additionally, we show that our proposed implicit ensemble extraction technique can recover epistemic uncertainty from large models.

Our work is of interest because epistemic uncertainty estimation is crucial for safety-critical applications and decision-making scenarios (Wen et al., 2021), where understanding model limitations can prevent costly errors. While the potential underestimation of epistemic uncertainty in large over-parameterized neural networks could lead to overconfident predictions and compromised decisions, our proposed implicit ensemble extraction technique offers a promising approach to recover and properly quantify this uncertainty.

In the remainder of this paper, we examine epistemic uncertainty collapse in ensembles of ensembles and in wider neural networks. We offer theoretical justification for the former and propose implicit ensembling as a potential mechanism in the latter. We provide empirical evidence of such a collapse in wider models through experiments with toy examples and MLPs trained on MNIST. Vice-versa, we show that *implicit ensemble extraction* methods, which decompose larger models into sub-models, recover hidden ensemble structure and epistemic uncertainty in both simple MLPs and state-of-the-art vision models based on ResNets (He et al., 2015) and Vision Transformers (Dosovitskiy et al., 2021).

## 2 Theoretical Framework

The quantification of uncertainty is crucial for robust and reliable predictions. We begin by distinguishing between two important types of uncertainty (Der Kiureghian & Ditlevsen, 2009):

1. **Aleatoric uncertainty**: Often referred to as statistical uncertainty, it represents the inherent noise in the data. This type of uncertainty is irreducible given the current set of features and cannot be reduced by collecting more data.
2. **Epistemic uncertainty**: Also known as model uncertainty, it stems from our lack of knowledge about the true data-generating process. This uncertainty can, in principle, be reduced by gathering more data or by using more expressive models.

**Traditional Neural Networks vs. Bayesian Neural Networks.** Traditional Neural Networks (NNs), while powerful function approximators, typically provide point estimates as outputs and inherently do not

capture model uncertainty. This limitation means that regular NNs cannot easily distinguish between aleatoric and epistemic uncertainty.

In contrast, Bayesian Neural Networks (BNNs) infer a distribution over model parameters instead of single-point estimates. This probabilistic approach allows BNNs to capture epistemic uncertainty. When a BNN encounters data similar to its training distribution, the predictive distributions across the parameter distribution tend to be more concentrated, indicating lower epistemic uncertainty. For out-of-distribution data, on the other hand,these distributions become more diffuse, indicating greater epistemic uncertainty.

**Bayesian Model Average.** The *Bayesian Model Average (BMA)* usually lies at the core of Bayesian deep learning. Let $p(\theta)$ be the distribution over model parameters $\Theta$. We are interested in the posterior distribution $p(\theta \mid \mathcal{D})$ given some observed data $\mathcal{D}$, but this is not necessary for the rest of this work. Thus, let us use $p(\theta)$ as the distribution of interest over model parameters: here, it will be the distribution of possible model parameters that we obtain by running a training procedure given the training data and different random initializations.

Let $Y$ be the predicted output with likelihood $p(y \mid \mathbf{x}, \theta)$. Then, the predictive distribution for a new input $\mathbf{x}^*$ is:

$$p(y^* \mid \mathbf{x}^*) = \int p(y^* \mid \mathbf{x}^*, \theta)\, p(\theta)\, d\theta. \tag{1}$$

**Information-Theoretic Quantities.** To formally quantify and differentiate between aleatoric and epistemic uncertainty, we can use an information-theoretic decomposition (Smith & Gal, 2018). Let $Y$ be the predicted output, and $\theta$ be the model parameters. We define:

1. **Total Uncertainty** as the *entropy* of the predictive distribution of the BMA:

$$H[Y \mid \mathbf{x}] = - \int p(Y \mid \mathbf{x}) \log p(Y \mid \mathbf{x}) dY. \tag{2}$$

2. **Epistemic Uncertainty** ($I[Y; \Theta \mid \mathbf{x}]$) as the *mutual information* which estimates the expected reduction in uncertainty about the prediction $Y$ that would be obtained if we knew the model parameters $\Theta$:

$$I[Y; \Theta \mid \mathbf{x}] = H[Y \mid \mathbf{x}] - \mathbb{E}_{p(\theta)}[H[Y \mid \mathbf{x}, \theta]]. \tag{3}$$

It has been shown to be an effective measure of epistemic uncertainty (Houlsby et al., 2011; Gal et al., 2017; Smith & Gal, 2018).

While the choice of uncertainty measures remains an active research area (Hüllermeier & Waegeman, 2021; Wimmer et al., 2023; Bengs et al., 2022; Shen et al., 2024a), mutual information has strong theoretical and empirical support. The mutual information between predicted outputs and model parameters directly captures the core idea of epistemic uncertainty: how much additional data collection could reduce our predictive uncertainty. This interpretation is used in both active learning (Houlsby et al., 2011; Gal et al., 2017) and Bayesian optimal experimental design (Rainforth et al., 2024), where it is known as the expected information gain. The effectiveness of this approach requires only that the training procedure is *approximately Bayesian* in its treatment of new data. This property has been demonstrated across many machine learning methods (Mlodozeniec et al., 2024; Kirsch, 2023). Recent studies confirm the strong empirical performance of this mutual information compared to alternative uncertainty measures (Schweighofer et al., 2024). Other approaches based on variance decomposition via the law of total variance have been proposed (Kendall & Gal, 2017); we briefly compare these methods in §A.1.

## 2.1 Deep Ensembles

Deep ensembles, introduced by Lakshminarayanan et al. (2017), have emerged as a simple yet effective method for approximating Bayesian inference in deep learning (Wilson & Izmailov, 2020). For a deeper discussion of the connection between deep ensembles and approximate Bayesian inference, see §A.2. They have been shown to offer superior performance compared to more complex Bayesian approximations on many tasks (Ovadia et al., 2019), including calibration measurements (Guo et al., 2017), out-of-distribution detection,

and classification with rejection (Ashukha et al., 2020). The core idea is to train multiple neural networks independently using different random initializations. The BMA of a deep ensemble of $M$ models is:

$$p(Y \mid \mathbf{x}) \approx \frac{1}{M} \sum_{j=1}^{M} p(Y \mid \mathbf{x}, \theta_j), \tag{4}$$

where $\theta_j \sim p(\theta_j)$ are the parameters of the $j$-th model in the ensemble, drawn from the distribution of possible trained models $p(\theta)$. Given uniform index distribution $p(j)$, we can estimate the epistemic uncertainty via the mutual information between the predicted class and the model index (Beluch et al., 2018):

$$I[Y; \Theta \mid \mathbf{x}] \approx H(\mathbb{E}_{p(j)}[p(Y \mid \mathbf{x}, \theta_j)]) - \mathbb{E}_{p(j)}[H[Y \mid \mathbf{x}, \theta_j]]. \tag{5}$$

This is just a Monte-Carlo approximation of the mutual information between the predicted class and the parameter distribution where the ensemble members are seen as samples from $p(\theta)$.

## 3 Epistemic Uncertainty Collapse for Ensembles of Ensembles

Let us formally investigate a paradoxical phenomenon: the collapse of epistemic uncertainty when constructing ensembles of ensembles. A standard deep ensemble comprises $M$ independently trained models that approximate a Bayesian Model Average (BMA). A different way to "scale up" deep ensembles is to create *multiple* deep ensembles and then average their predictions. One might expect that this *ensemble of ensembles* would yield more robust epistemic uncertainty estimates. However, our analysis has the opposite outcome: as we increase the size of each individual ensemble, the disagreement between these ensembles—and consequently, our estimate of epistemic uncertainty—vanishes.

### 3.1 Setup and Predictive Distribution

Let $S$ be the number of sub-ensembles, and let each sub-ensemble $\mathcal{E}_i$ contain $M$ models. Denote the parameters of the $j$-th model in the $i$-th sub-ensemble by $\theta_{i,j} \sim p(\theta)$. Each sub-ensemble provides a predictive distribution:

$$p(y \mid \mathbf{x}, \mathcal{E}_i) = \frac{1}{M} \sum_{j=1}^{M} p(y \mid \mathbf{x}, \theta_{i,j}). \tag{6}$$

Once we collect all sub-ensembles into $\mathcal{E}_{1:S} := \{\mathcal{E}_1, \ldots, \mathcal{E}_S\}$, the aggregated predictive distribution is

$$p(y \mid \mathbf{x}, \mathcal{E}_{1:S}) = \frac{1}{S} \sum_{i=1}^{S} p(y \mid \mathbf{x}, \mathcal{E}_i) \tag{7}$$

$$= \frac{1}{SM} \sum_{i=1}^{S} \sum_{j=1}^{M} p(y \mid \mathbf{x}, \theta_{i,j}). \tag{8}$$

Viewed as a single (larger) ensemble, this model is a mixture over all $S \times M$ members. Denote by $I \in \{1, \ldots, S\}$ the random variable for the choice of sub-ensemble, and by $J \in \{1, \ldots, M\}$ the random variable for the choice of model within that sub-ensemble. Then each $\theta_{i,j}$ is chosen with probability $\frac{1}{SM}$, yielding a Bayesian mixture over members. The empirical epistemic uncertainty within sub-ensemble $i$ is given by $I[Y; \theta_{i,J} \mid \mathbf{x}]$ and the empirical epistemic uncertainty across sub-ensembles is given by $I[Y; \mathcal{E}_I \mid \mathbf{x}]$.

### 3.2 Limit of Increasing Sub-Ensemble Size

A key observation is that growing $M$, holding $S$ fixed, causes all sub-ensembles to converge to identical predictive distributions, so any disagreement across sub-ensembles vanishes. Formally, for $M \to \infty$,

$$p(y \mid \mathbf{x}, \mathcal{E}_k) = \frac{1}{M} \sum_{i=1}^{M} p(y \mid \mathbf{x}, \theta_{i,j}) \to \mathbb{E}_{p(\theta)}[p(y \mid \mathbf{x}, \theta)], \tag{9}$$

since the LHS is a Monte-Carlo estimator of the RHS term. This implies that sub-ensemble indices $i$ become irrelevant: all sub-ensembles yield identical outputs in the limit. Consequently, the *epistemic uncertainty* of the ensemble of ensembles—the mutual information between the ensemble index ($I$) and the predicted label $Y$—collapses to zero as the predictions of $Y$ become independent of the ensemble index. In other words,

$$\mathrm{I}[Y; \mathcal{E}_I \mid \mathbf{x}] \to 0 \quad \text{as} \quad M \to \infty. \tag{10}$$

We call this *epistemic uncertainty collapse*.

> **Epistemic Uncertainty Collapse**
>
> As the size of the sub-ensemble in an ensemble of ensembles increases, the epistemic uncertainty of the overall ensemble approaches zero, and we observe an *epistemic uncertainty collapse*. This collapse occurs because the individual ensembles converge to similar predictive distributions, leading to a reduction in the mutual information between the predicted class and the ensemble index.

See §A.3 for a more detailed and formal argument.

### 3.3 Chain Rule Decomposition

To understand how the epistemic uncertainty of a single ensemble relates to that of an ensemble of ensembles, note that

$$\mathrm{I}[Y; \theta_{I,J} \mid \mathbf{x}] = \mathrm{I}[Y; \mathcal{E}_I \mid \mathbf{x}] + \mathrm{I}[Y; \theta_{I,J} \mid \mathcal{E}_I, \mathbf{x}]. \tag{11}$$

In words, the epistemic uncertainty of a large ensemble can be broken into two parts: disagreement *across* sub-ensembles $\mathrm{I}[Y; \mathcal{E}_I \mid \mathbf{x}]$ plus disagreement *within* each sub-ensemble $\mathrm{I}[Y; \theta_{I,J} \mid \mathcal{E}_I, \mathbf{x}]$. As $M \to \infty$, each sub-ensemble individually captures a near-complete BMA, so there is negligible disagreement across sub-ensembles.

## 4 Implicit Ensembling

Having established the epistemic uncertainty collapse for explicit ensembles of ensembles, this raises the question whether similar phenomena occur within individual large neural networks. We hypothesize that as neural networks increase in width, they might naturally exhibit ensembling internally, leading to reduced epistemic uncertainty estimates. This would have important implications for uncertainty quantification in large models, including modern foundation models.

Concretely, we propose that large neural networks inherently function as "*implicit ensembles*," where operations like matrix multiplication (or average pooling) effectively average across many units similar to averaging predictions in explicit ensembles. Under this hypothesis, wider networks would exhibit stronger internal ensembling effects, and ensembles of such networks would behave like ensembles of ensembles. This suggests that the epistemic uncertainty collapse demonstrated in §3.2 would manifest as model width increases, potentially limiting uncertainty quantification benefits from simply scaling model size.

This implicit ensembling hypothesis guides our empirical investigations in the subsequent section. We examine whether epistemic uncertainty decreases as MLP width increases despite stable accuracy, and whether we can extract ensemble-like structures from single large models to recover the collapsed uncertainty. We also investigate average pooling in vision models as another potential form of implicit ensembling, where activations from different receptive fields are aggregated in ways that might contribute to uncertainty collapse.

### 4.1 Connection to Neural Tangent Kernel Theory

The hypothesized epistemic uncertainty collapse in larger models connects elegantly with Neural Tangent Kernel (NTK) theory (Jacot et al., 2018; Lee et al., 2019; Yang, 2019). NTK theory characterizes infinitely wide neural networks trained with gradient descent, showing that under appropriate conditions (initialization and learning rate), these networks converge to Gaussian Process predictors determined by the NTK. That

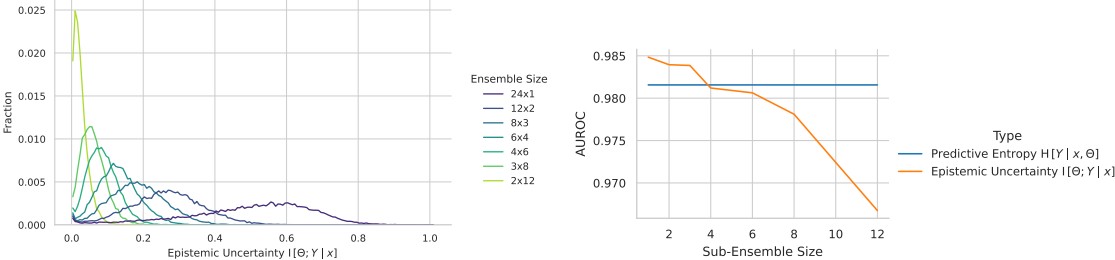

(a) Mutual Information Histogram on SVHN (OoD)   (b) OoD Detection for CIFAR10 vs. SVHN (OoD)

Figure 2: **Ensemble of Ensemble Results for CIFAR10 (iD) vs. SVHN (OoD).** Different configurations of 24 ResNet-50 models trained on CIFAR-10. *(a)* As the sub-ensemble size increases, the epistemic uncertainty on SVHN as OoD dataset collapses. *(b)* The area under the receiver-operating characteristic (AUROC ↑) for OoD detection using mutual information slowly deteriorates as the sub-ensemble size increases.

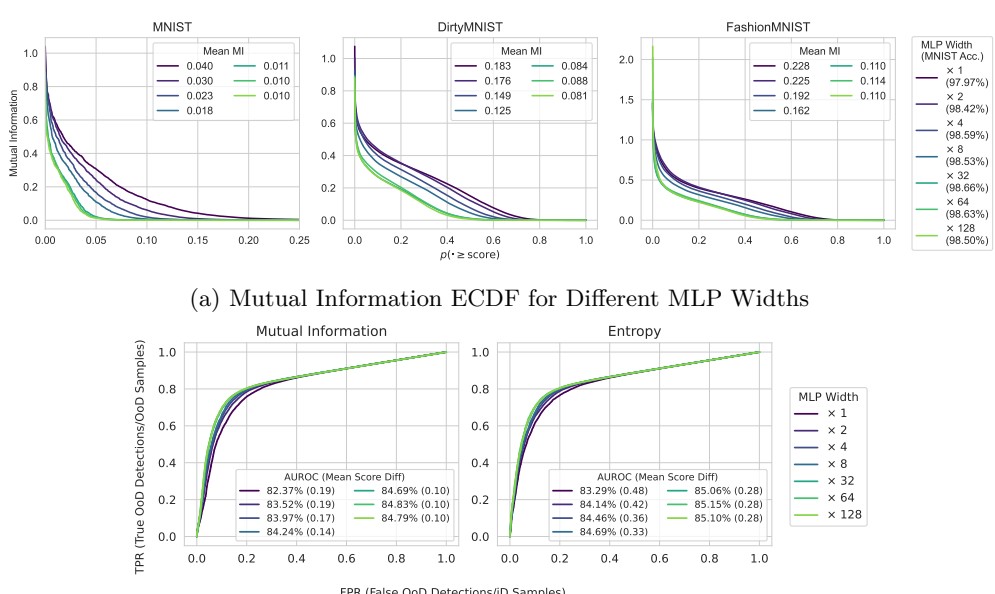

(a) Mutual Information ECDF for Different MLP Widths

(b) MNIST vs. FashionMNIST OoD Detection AUROC

Figure 3: **Epistemic Uncertainty Collapse on MNIST via Implicit Ensembling.** *(a) Mutual Information Empirical Cumulative Distribution Function (ECDF) for Different MLP Widths.* As MLP size increases, mutual information decreases while accuracy remains stable. This trend persists across training and other distributions. *(b) MNIST vs. Fashion-MNIST OoD Detection AUROC Curves.* The mean difference in uncertainty scores between in-distribution and out-of-distribution samples (in parentheses) also decreases with width, further evidencing epistemic uncertainty collapse, while the AUROC for OoD detection slightly improves across both uncertainty metrics.

is, for a fixed architecture, networks converge to identical predictive functions regardless of their random initialization in the infinite-width limit. Different initializations can follow different optimization paths but ultimately reach the *same* functional form. This has profound implications for uncertainty estimation.

In an ensemble of infinitely wide networks, each member converges to the same predictor function. The disagreement between ensemble members, which is the foundation of epistemic uncertainty estimation, therefore vanishes completely. Any uncertainty measure based on this disagreement, such as mutual information between predictions and ensemble membership, collapses to zero. This directly parallels our earlier findings

in §3.2: as sub-ensemble size increases, each sub-ensemble's predictive distribution approaches the common NTK-governed predictor, eliminating inter-ensemble disagreement.

Thus, NTK theory offers a theoretical framework that could be related to our proposed implicit ensembling hypothesis, though the connection remains speculative. As neural networks grow wider, their behavior increasingly approaches the NTK limit where all initializations converge to identical predictive distributions. We hypothesize that individual network components may still maintain considerable internal diversity, but this diversity might become progressively masked at the output level due to averaging effects in wider layers and pooling operations. If this hypothesis is correct, as width increases, such implicit averaging mechanisms would grow stronger, causing the network's collective behavior to approach the NTK-governed predictor despite diverse internal representations. While NTK characterizes the infinite-width limit where uncertainty vanishes at the output, the implicit ensembling framework might help explain how this phenomenon occurs in practical finite-width networks and suggest how hidden uncertainty could potentially be recovered by accessing these diverse internal components. We present initial evidence for this through implicit ensemble extraction in §5.1.

## 5 Empirical Results

In the following section, we present a series of experiments that demonstrate the epistemic uncertainty collapse in explicit ensembles of ensembles and show similar patterns in the behavior of wide neural networks. While these patterns are consistent with the implicit ensembling hypothesis we have proposed, we note that other explanations might also account for these observations. Details on the models, training setup, datasets, and evaluation are provided in §B and in §C in the appendix.

**Toy Example.** To illustrate the epistemic uncertainty collapse in ensembles of ensembles, we present a one-dimensional regression task with the ground-truth function $f(x) = \sin(x) + \epsilon$, where $\epsilon \sim \mathcal{N}(0, 0.1)$. We generate a small training dataset of four points over $[-5, 5]$ and fit this data using ensembles of neural networks with one hidden layer of 64 units and tanh activations. The output layer also uses a tanh activation, scaled by a factor of 2 to allow for a output range that surely covers any noised sample.

We create ensembles of 10 sub-ensembles each, with sub-ensemble sizes of 1, 2, 4, 8, 16, 32, and 64 members. Figure 1 presents the results across three panels. The left panel shows ensemble predictions converging to the true function as sub-ensemble size increases, with narrowing uncertainty bands. The middle panel illustrates decreasing epistemic uncertainty across the input space for larger sub-ensembles between training points. The right panel quantifies the inverse relationship between sub-ensemble size and mean epistemic uncertainty.

These results demonstrate the epistemic uncertainty collapse phenomenon: the systematic reduction of epistemic uncertainty, visible as reduced prediction variance and thus higher overall confidence, even in regions far from the training data. This toy example highlights how the proposed implicit ensembling within wide neural networks could lead to underestimating epistemic uncertainty, particularly in data-sparse regions.

In §A.4 in the appendix, we observe that epistemic uncertainty collapse also occurs in ensembles of random forests (Breiman, 2001), a fundamentally different model class from neural networks. As we discuss in §6, this distinguishes epistemic uncertainty collapse from phenomena like feature collapse (Van Amersfoort et al., 2020) and neural collapse (Papyan et al., 2020) that are specific to deep neural networks and their training dynamics.

**Explicit Ensemble of Ensemble.** In Figure 2, we construct a deep ensemble comprising of 24 Wide-ResNet-28-1 models (Zagoruyko & Komodakis, 2016; He et al., 2015) trained on CIFAR-10 (Krizhevsky et al., 2009), which we then partition into ensembles of ensembles with varying sub-ensemble sizes ($24 \times 1, 12 \times 2, 8 \times 3, 6 \times 4, 4 \times 6, 3 \times 8, 2 \times 12$). Figure 2(a) illustrates the epistemic uncertainty of these ensemble configurations. As the sub-ensemble size increases, we observe a clear epistemic uncertainty collapse, manifested by the mutual information concentrating on smaller values. Figure 2(b) presents the AUROC for out-of-distribution (OoD) detection for different sub-ensemble sizes, comparing CIFAR-10 (in-distribution) with SVHN (OoD) (Netzer et al., 2011) using epistemic uncertainty as the detection metric. The AUROC shows a deterioration as the sub-ensemble size increases, directly resulting from the epistemic uncertainty collapse. Crucially, while the decrease in AUROC may appear modest, it is large enough to make the difference between state-of-the-art

performance and baseline methods, such as confidence or entropy-based approaches (Hendrycks & Gimpel, 2017), e.g., compare with the results in Mukhoti et al. (2023).

**Implicit Ensembling on MNIST.** Surprisingly, the effect of the epistemic uncertainty collapse is even visible when training relatively small MLP models of varying width on MNIST in a controlled setting. We reproduce the results from Fellaji & Pennerath (2024) in Figure 3. That is, we expand the size of the inner linear of a 2-hidden-layer MLP by a varying factors and train 10-model ensembles for 100 epochs each.

To quantify uncertainty, we employ two metrics that capture different aspects of the model's predictive distribution. Specifically, we utilize: *(1)* The predictive entropy of the ensemble, $H[Y \mid \mathbf{x}]$, which measures the overall uncertainty in the prediction (Hendrycks & Gimpel, 2017); and *(2)* the mutual information between the predicted class and the ensemble index, as an empirical estimate for $I[Y; \Theta \mid \mathbf{x}]$, which quantifies the epistemic uncertainty or model uncertainty (Malinin, 2019).

These results demonstrate epistemic uncertainty collapse in wider neural networks. This pattern is consistent with our implicit ensembling hypothesis, though other explanations may also be possible. As the width of the MLP increases, we observe a clear trend of decreasing mutual information across all datasets: MNIST (LeCun & Cortes, 1998), Dirty-MNIST (Mukhoti et al., 2023), and Fashion-MNIST (Xiao et al., 2017). In Figure 3(a), the effect is most pronounced for the in-distribution MNIST test data, but the collapse is also clearly visible for out-of-distribution data (Fashion-MNIST & SVHN test sets). The decrease in mutual information indicates a reduction in the model's epistemic uncertainty as it grows larger, despite maintaining similar accuracy. In Figure 3(b), the mean difference in uncertainty scores between in-distribution (MNIST) and out-of-distribution (Fashion-MNIST) samples also decreases with increasing model width, further corroborating the collapse of epistemic uncertainty. However, the AUROC for OoD detection using different uncertainty metrics slightly improves as the model width increases, which is in line with the results by Fellaji & Pennerath (2024), who report a significant deterioration of OoD performance for such MLP ensembles trained on CIFAR-10 but not MNIST.

This shows that epistemic uncertainty collapse might be a pervasive phenomenon in neural networks, even in relatively simple MLP models.

## 5.1 Implicit Ensemble Extraction

To further investigate the hypothesis that implicit ensembling might contribute to epistemic uncertainty collapse, we demonstrate that it is possible to mitigate this collapse by decomposing larger models into constituent sub-models. This approach, which we term implicit ensemble extraction, allows us to recover the diversity of an ensemble from a single large model, potentially reversing the collapse of epistemic uncertainty. This not only offers insights into the nature of epistemic uncertainty collapse but also points towards potential practical strategies for improving model robustness and out-of-distribution detection.

**Extracted Implicit Ensemble from a Single MNIST MLP.** First, we extract implicit ensembles from the MLPs trained on MNIST above. For a given model, an ensemble member with the largest width factor, 64, we train boolean masks on the weights such that we recover 10 individual models with maximally different masks and low individual loss on MNIST's training set. We find that the resulting deep ensemble performs as well as the ensemble of smaller models, even though we started from a single model.

Concretely, we add binary mask to each linear layer of the network which we relax to probabilities of binomial variables by applying sigmoid activations, effectively selecting subsets of the original weights. We optimize separate 1D masks for its rows and columns. The outer product of these 1D masks determines which weights from the original layer are included in each sub-model as a dense sub-matrix. We maximize the diversity among the resulting sub-models by regularizing the mutual information $I[\text{mask}; M]$ between the masks and sub-model index $M$ while minimizing the loss on the training set. This allows us to extract an implicit ensemble from a single trained network.

In Figure 4, we see the effectiveness of this implicit ensemble extraction technique. The results demonstrate that this decomposition can recover much of the epistemic uncertainty of an ensemble from a single model, providing support for our hypothesis about implicit ensembling:

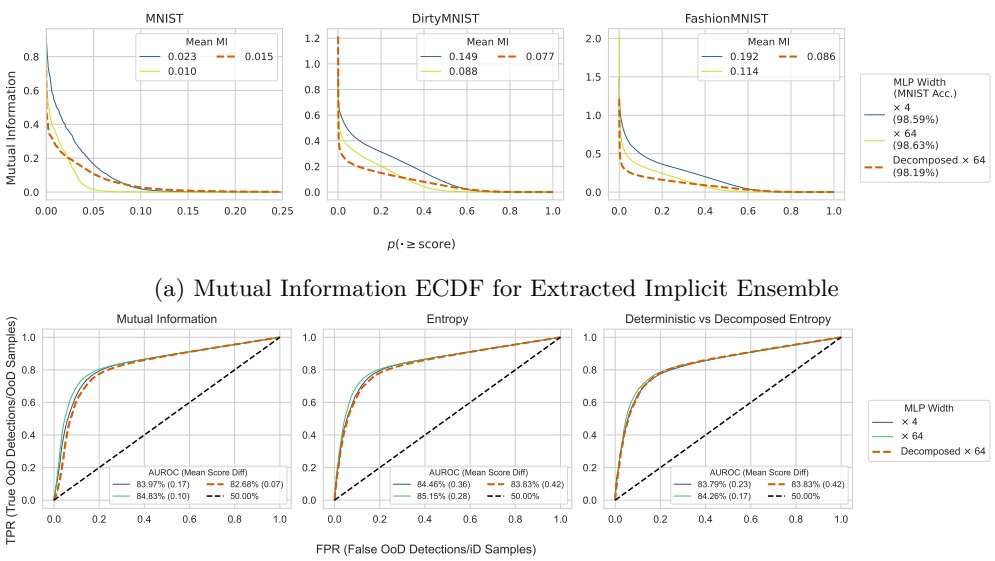

(a) Mutual Information ECDF for Extracted Implicit Ensemble

(b) MNIST vs. FashionMNIST OoD Detection AUROC for Extracted Implicit Ensemble

Figure 4: **Recovering Epistemic Uncertainty through Implicit Ensemble Extraction.** *(a)* The extracted implicit ensemble (dashed line) largely recovers the mutual information scores of a fully trained ensemble of the same width, supporting the hypothesis of latent ensemble structures in large neural networks. The 10-member implicit ensemble is extracted from a single MLP with width factor 64 (§5.1). The regular 10-member ensembles comprise MLPs with width factors 4 and 64 trained on MNIST. Ensembles are evaluated on MNIST, Dirty-MNIST, and Fashion-MNIST test sets. *(b)* The extracted implicit ensemble shows comparable AUROC scores across all metrics relative to a fully trained deep ensemble of the same width. OoD detection is performed using mutual information or entropy scores. The final panel compares the softmax entropy of the original wide MLP with the predictive entropy of its extracted implicit ensemble. The mean entropy difference between iD and OoD samples is larger for the extracted ensemble. At the same time, the OoD performance does *not* match the single wider MLP.

- **Effectiveness of Model Decomposition.** The dashed lines in both subfigures of Figure 4 represent the extracted ensemble, as described in the previous section. Remarkably, this decomposed model nearly recovers the mutual information scores of a fully trained ensemble with the same width factor despite being based on the weights of a single model, and its OoD detection performance approaches that of a narrower model. The fact that we can successfully extract an ensemble structure suggests that diverse functional components may exist within larger models, which is consistent with our implicit ensembling hypothesis.
- **Predictive Entropy of the Extracted Ensemble vs. the Softmax Entropy of the Original Model.** The last panel of Figure 4(b) compares the softmax entropy of the original large model with the predictive entropy of the decomposed ensemble. While we cannot compare the OoD detection performance of the extracted ensemble to the single wider MLP using mutual information (as the latter does not have a well-defined mutual information), we see that the extracted ensemble does not match the performance of the single wider MLP when using its predictive entropy. We speculate this might be because the masks do not fully cover the weights of the original model, and thus the extracted ensemble does not capture the full predictive entropy of the original ensemble.

The initial success of this model decomposition technique in recovering epistemic uncertainty suggest a promising direction for improving uncertainty quantification and out-of-distribution detection in individual over-parameterized deep learning models. Collectively, this demonstrates that epistemic uncertainty collapse occurs even in relatively simple MLP models.

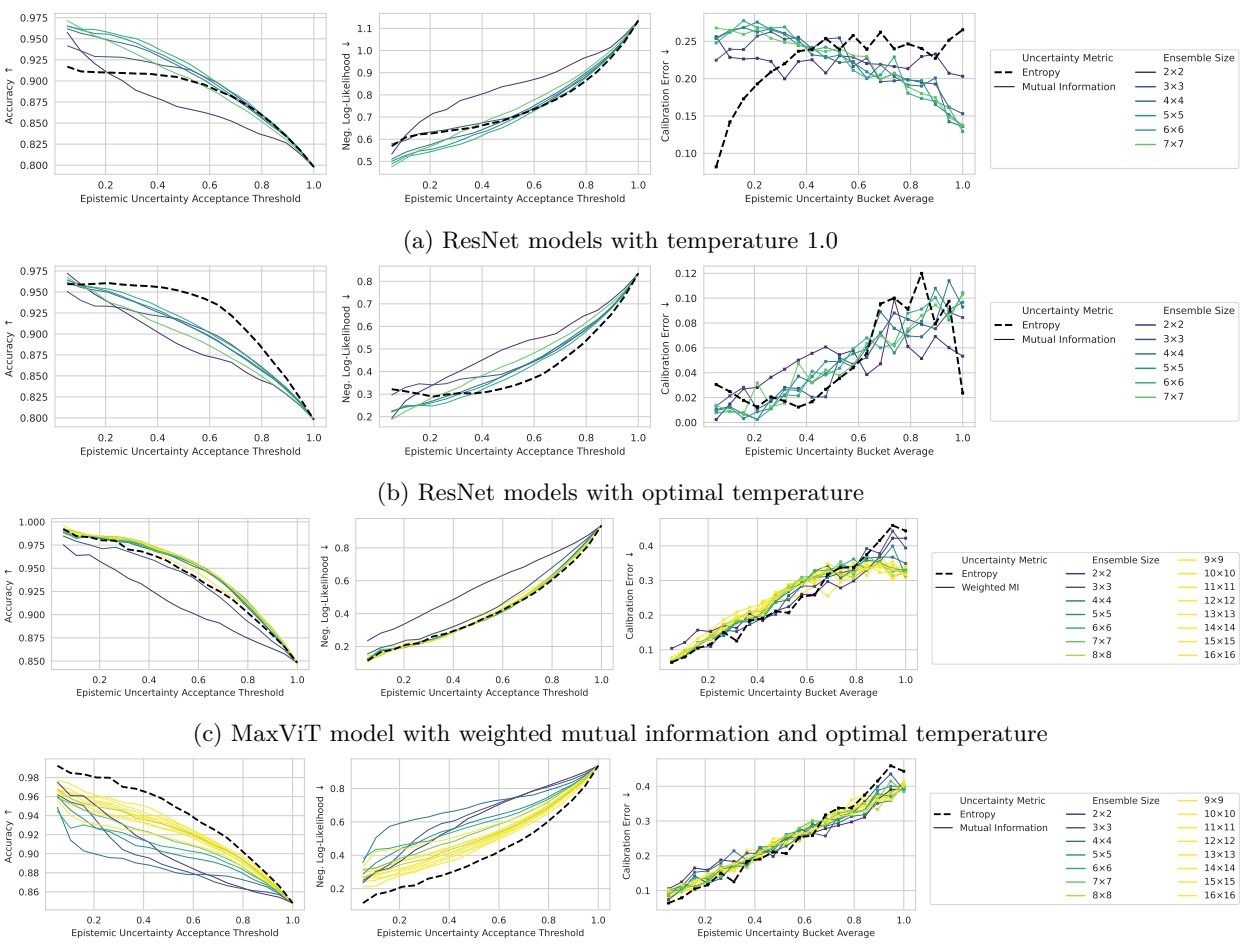

(a) ResNet models with temperature 1.0

(b) ResNet models with optimal temperature

(c) MaxViT model with weighted mutual information and optimal temperature

(d) MaxViT model with mutual information and optimal temperature

Figure 5: **Classification with Rejection for Implicit Ensemble Extractions from Pre-Trained Models.** Each subfigure shows three performance metrics (Accuracy, Negative Log-Likelihood, and Calibration Error) as a function of epistemic uncertainty quantiles for different ensemble sizes. Solid lines represent extracted ensembles of increasing size (from 2 to 7/16), while the dashed black line represents the original single model. *(a)* The mutual information between predictions is used as the epistemic uncertainty measure for ensembles, while entropy is used for the single model. As the ensemble size increases, we observe improved performance for the area under curve (AUC), which indicates better epistemic uncertainty calibration (with the notable exception of the calibration error). This demonstrates that extracting larger ensembles from a single pre-trained model can enhance performance and uncertainty quantification. *(b)* Temperature scaling improves epistemic uncertainty calibration in general but benefits the original model most. Accuracy and NLL for extracted epistemic uncertainty only benefit in the low-uncertainty regime. *(c)* For VIT models, we find that a mutual information weighted by the logit sum of each ensemble performs better than the mutual information (*(c)* vs *(d)* with mutual information).

**Extracting Ensembles from Pre-Trained Vision Models.** Second, we explore implicit ensemble extraction from pre-trained vision models based on ResNet (He et al., 2015) and Vision Transformer (Dosovitskiy et al., 2021) model architectures. Leveraging the common use of average pooling in these models to aggregate spatial information, we extract implicit ensembles without optimizing masks. Concretely, we remove the global average pooling layer. This allows us to obtain per-tile class logits, which we average with different target sizes to create differently-sized ensembles.

We evaluate these models in-distribution on the ImageNet-v2 dataset (Recht et al., 2019), which serves as a more challenging test set for ImageNet-trained models (Russakovsky et al., 2015). Specifically, we compare pre-trained ResNet-152 (He et al., 2015), Wide ResNet-101-2 (Zagoruyko & Komodakis, 2016), ResNeXt-101-64x4d (Xie et al., 2017), and MaxViT (Tu et al., 2022) models, with the pre-trained weights retrieved from PyTorch's torchvision (maintainers & contributors, 2016) and timm (Wightman, 2019), respectively. Our evaluation pipeline computes various uncertainty metrics and performance measures for different ensemble sizes, ranging from $2 \times 2$ to $7 \times 7$ sub-models (respectively, $16 \times 16$ for MaxViT) extracted from a single pre-trained network. We compare these extracted ensembles against the original single model performance, using mutual information as the primary uncertainty metric for ensembles and entropy for the single model.

Figure 5 shows three key performance metrics—accuracy, negative log-likelihood (NLL), and calibration error—plotted against epistemic uncertainty quantiles of various extracted ensemble sizes for the original models and for temperature-scaled models (Guo et al., 2017) The solid lines represent extracted ensembles of increasing size, while the dashed black line represents the original single model. For ResNet-based ensembles, we use mutual information between predictions as the measure of epistemic uncertainty, whereas for the single model, we use entropy. For MaxViT, we use a weighted mutual information between predictions and ensemble size as the measure of epistemic uncertainty, which assign a weight to each ensemble member based on the logit sum of the member as it performs better.

For ResNet models, we find that the largest extracted ensembles (7x7) have a better response overall for accuracy and NLL, while the calibration error is worse throughout. For temperature-scaled model, we observe an almost linear response in accuracy and NLL, with better sensitivity in the low-uncertainty regime, but otherwise worse performance, except for a lower calibration error. For MaxViT, we find that the weighted mutual information performs better than the mutual information. The effects are less pronounced for temperature-scaled models, likely because these models already are better calibrated. Overall, the results are mixed but promising.

**Limitations.** These *exploratory* results provide initial evidence that ensemble extraction technique can unlock meaningful epistemic uncertainty, which is useful for downstream tasks. While this is consistent with the implicit ensembling hypothesis, it does not prove that implicit ensembling is the mechanism responsible for uncertainty collapse in wide networks. At the same time, we already see that mutual information is not always the best uncertainty metric, the comparative performance changes depending on the model temperature, and the pool size of the best implicit ensemble is not always the same for different metrics and model architectures. More results that show this are shown in §E.

## 6 Related Work

Our study of epistemic uncertainty collapse in large neural networks and ensemble extraction intersects with several recent works in adjacent areas.

**Epistemic Uncertainty Collapse.** A recent preprint by Fellaji & Pennerath (2024) observed decreasing epistemic uncertainty as model size and dataset size vary, even in simple MLPs. They termed the decrease in epistemic uncertainty the "epistemic uncertainty hole" but left its explanation to future work. Our study proposes an explanation through a theoretical framework for ensembles of ensembles, the hypothesis of implicit ensembling for wider models, and additional empirical evidence from experiments on ensemble extraction. A more detailed comparison can be found in §D.

**Extracting Sub-Models from a Larger Network.** The concept of extracting sub-models from a larger network shares similarities with several existing approaches in the literature. The "lottery ticket hypothesis" (Frankle & Carbin, 2018) proposes that dense, randomly-initialized networks contain sparse subnetworks capable of training to similar accuracy. However, our approach differs in that we do not retrain the subnetworks, but rather identify diverse substructures within the pre-trained model. Our method is more closely related to "sub-network ensembles" (Durasov et al., 2021), where multiple subnetworks are extracted from a single trained network to form an ensemble. Unlike previous work that primarily focused on pruning for efficiency, our approach aims to recover epistemic uncertainty. We introduce a novel mutual information-based objective to obtain diverse masks, emphasizing diversity rather than pruning. This allows us to extract an ensemble

that better captures the model's internal epistemic uncertainty. This goal is more similar to Wortsman et al. (2020) which is concerned with discovering sub-networks during training for continual learning of diverse tasks without catastrophic forgetting. Our approach is only applied post-training, however.

**Learning from Underspecified Data.** Our work can also related to efforts in learning from underspecified data, such as the approach by Lee et al. (2022) to diversify and disambiguate model predictions. While their focus is on training strategies, our method extracts diverse sub-models from already trained networks, offering a complementary approach.

In contrast to these works, this work addresses fundamentally different research questions. We propose a novel, unified explanation for epistemic uncertainty collapse in large models and hierarchical ensembles, a phenomenon not previously explored in depth. Furthermore, we introduce and examine implicit ensemble extraction to mitigate epistemic uncertainty collapse.

**Relationship to Feature and Neural Collapse.** Our findings on epistemic uncertainty collapse share interesting connections with, but remain distinct from, two related phenomena in deep learning. Feature collapse, introduced by Van Amersfoort et al. (2020), describes how intermediate feature representations become increasingly similar during training, leading to overconfident predictions. Neural collapse (Papyan et al., 2020) characterizes how features and classifiers converge to a specific geometric structure in the terminal phase of training.

While these phenomena involve forms of representation collapse, our observed epistemic uncertainty collapse operates at a different level. Rather than focusing on feature geometry or representation similarity, we study how predictive uncertainty estimates degrade in two key settings: (1) ensembles of ensembles, where individual ensembles converge to similar predictive distributions despite diverse training, and (2) large models through the proposed implicit ensembling, where we can extract diverse sub-models that reveal hidden uncertainty structure. The distinction becomes particularly clear in regression tasks (Figure 1), where neural collapse theory does not apply due to its reliance on classification-specific geometric properties.

## 7 Conclusion

This work has analyzed the collapse of epistemic uncertainty in large neural networks and hierarchical ensembles. Our theoretical framework and empirical results demonstrate this phenomenon across various architectures and datasets, from simple MLPs to state-of-the-art vision models.

To further explore this phenomenon, we examined implicit ensemble extraction, a method to decompose larger models into sub-models. This approach can recover diverse predictive distributions from a single model and improve epistemic uncertainty estimates. The effectiveness of this extraction method is consistent with our implicit ensembling hypothesis, though it does not definitively establish it as the underlying mechanism of epistemic uncertainty collapse.

Our findings challenge the assumption that more complex models invariably offer better uncertainty quantification out of the box. They open new avenues for research into uncertainty estimation in large-scale machine learning models, particularly for safety-critical applications and out-of-distribution detection tasks. Future work should investigate the exact mechanisms behind epistemic uncertainty collapse in large models and develop robust methods to maintain reliable uncertainty estimates as model complexity increases, ensuring the continued advancement of trustworthy AI systems.

**Acknowledgments**

We thank the anonymous TMLR reviewers and Freddie Bickford Smith for their helpful feedback. Alex Evans provided valuable comments on an earlier draft, and Midjourney generously supported this personal research last summer. Jishnu Mukhoti trained the model checkpoints used in the initial explicit ensemble of ensemble experiments on CIFAR-10 as part of Mukhoti et al. (2023). Discussions with Lisa Schut helped shape the intuitions about NTK and implicit ensembling. We also appreciate Jannik Kossen and Tom Rainforth's insights following an early presentation at OATML.

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

## A Theoretical Framework

### A.1 Variance-Based Epistemic Uncertainty

A convenient way to obtain a tractable upper bound for the mutual information $I[Y; \Theta \mid \mathbf{x}]$ uses the law of total variance on the random variable $Y$. Recall the law of total variance:

$$\mathrm{Var}[Y \mid \mathbf{x}] = \mathbb{E}[\mathrm{Var}[Y \mid \mathbf{x}, \Theta]] + \mathrm{Var}[\mathbb{E}[Y \mid \mathbf{x}, \Theta]]. \tag{12}$$

It is sometimes used as an uncertainty decomposition in its own right (Kendall & Gal, 2017), where the first term on the right $\mathbb{E}[\mathrm{Var}[Y \mid \mathbf{x}, \Theta]]$ is the average conditional variance across models, considered the aleatoric part, while the second term $\mathrm{Var}[\mathbb{E}[Y \mid \mathbf{x}, \Theta]]$ quantifies how the conditional mean changes with $\Theta$, considered the epistemic part.

To connect this decomposition to $I[Y; \Theta \mid \mathbf{x}]$, assume each conditional distribution $p(Y \mid \mathbf{x}, \Theta)$ is (approximately) Gaussian with constant covariance, and approximate $p(Y \mid \mathbf{x})$ using a Gaussian as maximum entropy distribution. The differential entropy of a Gaussian random variable with variance $\sigma^2$ is $\frac{1}{2} \log(2\pi e \, \sigma^2)$. This allows us to approximate $I[Y; \Theta \mid \mathbf{x}]$ by comparing the entropy of these Gaussians:

$$I[Y; \Theta \mid \mathbf{x}] = H[Y \mid \mathbf{x}] - H[Y \mid \mathbf{x}, \Theta] \tag{13}$$

$$= H[Y \mid \mathbf{x}] - \mathbb{E}_{p(\theta)}[H[Y \mid \mathbf{x}, \theta]] \tag{14}$$

$$\approx \frac{1}{2} \log(2\pi e \, \mathrm{Var}[Y \mid \mathbf{x}]) - \mathbb{E}_{p(\theta)}[\frac{1}{2} \log(2\pi e \, \mathrm{Var}[Y \mid \mathbf{x}, \theta])] \tag{15}$$

$$\leq \frac{1}{2} \log\left( \frac{\mathrm{Var}[Y \mid \mathbf{x}]}{\mathbb{E}[\mathrm{Var}[Y \mid \mathbf{x}, \Theta]]} \right) \tag{16}$$

$$= \frac{1}{2} \log\left( 1 + \frac{\mathrm{Var}[\mathbb{E}[Y \mid \mathbf{x}, \Theta]]}{\mathbb{E}[\mathrm{Var}[Y \mid \mathbf{x}, \theta]]} \right). \tag{17}$$

The numerator reflects epistemic variability (how much the mean shifts between models), and the denominator captures the typical conditional variance. This ratio then appears within a logarithm because differential entropy grows with the log of variance in the Gaussian setting.

From this, we can see that variance-based epistemic uncertainty can be loosely matched to mutual information when each conditional distribution $p(y \mid \mathbf{x}, \theta)$ is close to Gaussian with roughly constant variance, making total variance reflect the same factors that drive the mutual information. Outside this regime—particularly with multimodal, heavy-tailed, or strongly heteroscedastic predictive distributions—the variance-based approach masks richer dependencies in $p(y \mid \mathbf{x}, \theta)$, and the two metrics diverge as mutual information accounts for more complex, higher-order effects[2].

### A.2 Deep Ensembles as Approximate Posterior Sampling

The theoretical justification for using deep ensembles to estimate epistemic uncertainty stems from their connection to Bayesian inference (Wilson & Izmailov, 2020). When training neural networks with different random initializations and stochastic optimization, the resulting ensemble can be viewed as drawing approximate samples from the posterior distribution over model parameters (Stephan et al., 2017; Shen et al., 2024b).

This interpretation arises because the training process combines:

- Random initialization, which provides diverse starting points in parameter space (Garipov et al., 2018)
- Stochastic gradient descent, which introduces noise in optimization trajectories (Stephan et al., 2017)
- Weight decay regularization, which acts as an implicit prior over parameters (Loshchilov & Hutter, 2019)

---

[2]This section is based on an informal note from 2022: https://blackhc.notion.site/Law-of-Total-Variance-vs-Information-Theory-1608b02974c54839a7c4a8e181f75e7f.

While deep ensembles do not perform exact Bayesian inference, they capture important aspects of posterior uncertainty through the diversity of solutions found (Fort et al., 2019). The independent training runs explore different modes of the loss landscape (Garipov et al., 2018), approximating the kind of parameter diversity we would expect from true posterior samples. This provides a practical foundation for using ensembles to estimate epistemic uncertainty, even without explicit variational approximations or MCMC sampling (Wilson & Izmailov, 2020).

The quality of this approximation improves with ensemble size, as more independent training runs better characterize the range of plausible models given the data (Lakshminarayanan et al., 2017). This theoretical connection motivates why deep ensembles often provide better uncertainty estimates than methods that make stronger assumptions about the form of the approximate posterior (Ovadia et al., 2019).

### A.3 Epistemic Uncertainty Collapse in Ensembles of Ensembles

The rather informal exposition in §3.2 does not differentiate between empirical mutual information which measures the disagreement between the predictions of the ensemble members and the mutual information between the predictions and the parameter distribution where the ensemble members are seen as samples from $p(\theta)$. It is not necessary to understand the effect and the empirical results. Let us make this distinction clearer here for completeness' sake.

The empirical epistemic uncertainty of a specific sub-ensemble $\mathcal{E}_i$ is $I[Y; \theta_{I,J} \mid \mathcal{E}_i, \mathbf{x}]$. The epistemic uncertainty of the paramter distribution itself is $I[Y; \Theta \mid \mathbf{x}]$. We have:

$$I[Y; \theta_{I,J} \mid \mathcal{E}_i, \mathbf{x}] = H(\mathbb{E}_j[p(Y \mid \mathbf{x}, \theta_{i,j})]) - \mathbb{E}_j[H(p(Y \mid \mathbf{x}, \theta_{i,j}))], \tag{18}$$

where $p(j)$ follows a uniform distribution. We can write this more explicitly as:

$$I[Y; \theta_{I,J} \mid \mathcal{E}_i, \mathbf{x}] = H(\frac{1}{M} \sum_{j=1}^{M} p(Y \mid \mathbf{x}, \theta_{i,j})) - \frac{1}{M} \sum_{j=1}^{M} H(p(Y \mid \mathbf{x}, \theta_{i,j})). \tag{19}$$

The entropy is a continuous function, so we can take the limit as the the number of models in the sub-ensemble $M$ goes to infinity and swap the order of limit and entropy, where we use that $\frac{1}{M} \sum_{i=1}^{M} p(y|\mathbf{x}, \theta_{i,j}) \to \mathbb{E}_{p(\theta)}[p(y| \mathbf{x}, \theta)]$ following (9) and similarly $\frac{1}{M} \sum_{j=1}^{M} H(p(Y \mid \mathbf{x}, \theta_{i,j})) \to H(\mathbb{E}_{p(\theta)}[p(y \mid \mathbf{x}, \theta)])$. That is, for $M \to \infty$, we have:

$$I[Y; \theta_{I,J} \mid \mathcal{E}_i, \mathbf{x}] = H(\frac{1}{M} \sum_{j=1}^{M} p(Y \mid \mathbf{x}, \theta_{i,j})) - \frac{1}{M} \sum_{j=1}^{M} H(p(Y \mid \mathbf{x}, \theta_{i,j})) \tag{20}$$

$$\to H(\mathbb{E}_{p(\theta)}[p(y \mid \mathbf{x}, \theta)]) - \mathbb{E}_{p(\theta)}[H(p(y \mid \mathbf{x}, \theta))] \tag{21}$$

$$= H(p(y \mid \mathbf{x})) - \mathbb{E}_{p(\theta)}[H(p(y \mid \mathbf{x}, \theta))] \tag{22}$$

$$= I[Y; \Theta \mid \mathbf{x}]. \tag{23}$$

Now, crucially, we can make the same argument for the epistemic uncertainty of the ensemble comprising all models in all the sub-ensembles $I[Y; \theta_{I,J} \mid \mathbf{x}]$, and we obtain the same result:

$$I[Y; \theta_{I,J} \mid \mathbf{x}] \to I[Y; \Theta \mid \mathbf{x}]. \tag{24}$$

With the chain rule of mutual information(11) we obtain:

$$I[Y; \theta_{I,J} \mid \mathbf{x}] = I[Y; \mathcal{E}_I \mid \mathbf{x}] + I[Y; \theta_{I,J} \mid \mathcal{E}_I, \mathbf{x}]. \tag{25}$$

and in the limit of infinitely-sized sub-ensembles, the epistemic uncertainty across the sub-ensembles is thus sandwiched to 0:

$$I[Y; \mathcal{E}_I \mid \mathbf{x}] = I[Y; \theta_{I,J} \mid \mathbf{x}] - I[Y; \theta_{I,J} \mid \mathcal{E}_i, \mathbf{x}] \to I[Y; \Theta \mid \mathbf{x}] - I[Y; \Theta \mid \mathbf{x}] = 0. \tag{26}$$

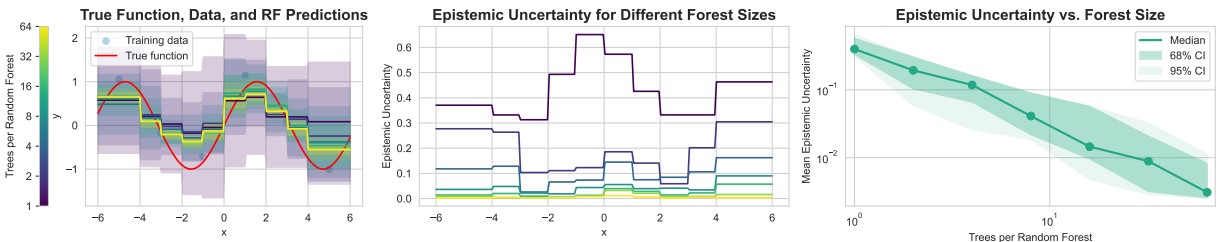

Figure 6: **Epistemic Uncertainty Collapse in Random Forest Ensembles.** The panels show the collapse of epistemic uncertainty as the number of trees per forest increases. *Left:* Model predictions and uncertainty bands ($\pm 2$ standard deviations) for different random forest sizes, showing convergence to the true function with reduced uncertainty as the random-forest ensemble size grows. *Middle:* Epistemic uncertainty across the input space for different random-forest ensemble sizes, demonstrating systematic reduction in epistemic uncertainty between as the random-forest ensemble size increases. *Right:* Relationship between random-forest ensemble size and mean epistemic uncertainty, with shaded regions showing 68% and 95% confidence intervals, revealing a clear inverse relationship.

### A.4 Epistemic Uncertainty Collapse in Random Forests

To further validate the epistemic uncertainty collapse phenomenon, we examine its manifestation in random forests—another ensemble learning method with well-understood theoretical properties. An ensemble of random forests (Breiman, 2001) naturally forms an ensemble of ensembles, where each random forest is an ensemble of decision trees.

We replicate the toy regression problem from earlier, training random forest ensembles with varying numbers of trees (1, 2, 4, 8, 16, 32, and 64) and creating ensembles of ensembles of 10 random forests for each configuration. Each random forest uses decision trees with a maximum depth of 3 to prevent overfitting while maintaining sufficient model capacity.

The results in Figure 6 mirror our findings with neural networks. As the number of trees in each random forest increases:

- the ensemble predictions converge more tightly to the true function, with narrowing uncertainty bands;
- the epistemic uncertainty decreases systematically across the input space, particularly in regions between training points; and
- the mean epistemic uncertainty shows a clear inverse relationship with forest size

This parallel between random forests and neural networks underscores that epistemic uncertainty collapse is a general phenomenon in ensemble methods, rather than being specific to neural networks. The observation that this effect appears in random forests—which have stronger theoretical guarantees and more interpretable behavior than neural networks—provides additional evidence for the fundamental nature of this phenomenon.

## B Model and Dataset Details

### B.1 MNIST Experiments

For the MNIST experiments, we use a simple Multi-Layer Perceptron (MLP) architecture with two hidden layers. The model structure is as follows:

- Input layer: 784 units (28x28 flattened MNIST images)
- First hidden layer: 64 units multiplied by a width multiplier (ranging from 0.5x to 256x)
- Second hidden layer: 32 units multiplied by the same width multiplier
- Output layer: 10 units (one for each digit class)
- Activation function: ReLU after each hidden layer

- Dropout layers: Applied after each hidden layer with p=0.1

We train these models using the following configuration:

- Optimizer: SGD
- Learning rate: 0.01
- Batch size: 128
- Epochs: 100
- Loss function: Cross-entropy loss

We create ensembles of 10 models for each width configuration. The width multipliers used are 1x, 2x, 4x, 8x, 32x, 64x, and 128x the base width.

Datasets used:

- MNIST (LeCun & Cortes, 1998): Standard handwritten digit dataset (in-distribution)
- Fashion-MNIST (Xiao et al., 2017): Clothing item dataset (out-of-distribution)
- Dirty-MNIST (Mukhoti et al., 2023): MNIST with added noise (high aleatoric uncertainty)

### B.2   CIFAR-10 Experiments

For the CIFAR-10 experiments, we use Wide-ResNet-28-1 models (Zagoruyko & Komodakis, 2016). We create an ensemble of 24 independently trained models, which we then partition into sub-ensembles of various sizes, following the training details of (Mukhoti et al., 2023).

Training configuration:

- Optimizer: SGD with momentum (0.9)
- Learning rate: 0.1, decayed by a factor of 10 at epochs 150 and 250
- Weight decay: 5e-4
- Batch size: 128
- Epochs: 350
- Loss function: Cross-entropy loss

Datasets used:

- CIFAR-10 (Krizhevsky et al., 2009): 10-class image classification dataset (in-distribution)
- SVHN (Netzer et al., 2011): Street View House Numbers dataset (out-of-distribution)

### B.3   ImageNet Experiments

For the ImageNet experiments, we use pre-trained models from the torchvision and timm libraries:

- ResNet-152 (He et al., 2015)
- Wide ResNet-101-2 (Zagoruyko & Komodakis, 2016)
- ResNeXt-101-64x4d (Xie et al., 2017)
- MaxViT (Tu et al., 2022)

These models are evaluated on the ImageNet-v2 dataset (Recht et al., 2019), which serves as a more challenging test set for ImageNet-trained models.

### B.4   Implicit Ensemble Extraction

For the implicit ensemble extraction experiments on MNIST, we use the following approach:

- Starting model: MLP with width factor 64
- Extraction method: Optimizing binary masks for each layer
- Number of extracted sub-models: 10
- Optimization objective: Maximize mask diversity (mutual information between masks and sub-model index) while minimizing the cross-entropy loss on training set

- Mask diversity weight: 2.0

For the pre-trained vision models, we extract implicit ensembles by:

- Removing the global average pooling layer
- Obtaining per-tile class logits
- Averaging these logits with different target sizes (from 2x2 to 7x7 for ResNets, and up to 16x16 for MaxViT)

Table 1: AUROC and Mean Difference for Different Metrics and MLP Widths

| | AUROC | | Mean MI Difference | |
| OoD Metric | MI | Entropy | MI | Entropy |
| MLP Width | | | | |
| --- | --- | --- | --- | --- |
| ×**1** | 0.824 | 0.833 | 0.188 | 0.482 |
| ×**2** | 0.835 | 0.841 | 0.194 | 0.420 |
| ×**4** | 0.840 | 0.845 | 0.169 | 0.362 |
| ×**8** | 0.842 | 0.847 | 0.144 | 0.329 |
| ×**32** | 0.847 | 0.851 | 0.099 | 0.285 |
| ×**64** | 0.848 | 0.851 | 0.104 | 0.280 |
| ×**1.3e+02** | 0.848 | 0.851 | 0.100 | 0.275 |

## C Evaluation Details

This section provides detailed information about our evaluation metrics, with a particular focus on the weighted mutual information and calibration error calculations.

### C.1 Weighted Mutual Information

For the MaxViT model, we introduce a weighted mutual information metric to measure epistemic uncertainty. This metric assigns a weight to each ensemble member based on the logit sum of that member. The weighted mutual information is calculated as follows:

1. For each input, we compute the logits for all ensemble members.
2. We calculate the sum of logits for each ensemble member.
3. We normalize these sums to create weights for each member.
4. We compute the mutual information between the predictions and the ensemble index, weighting each member's contribution by its normalized logit sum.

Formally, let $l_{i,j}$ be the logit sum for the $i$-th input and $j$-th ensemble member. The weight $w_j$ for this member is:

$$w_j = \frac{\sum_i \exp(l_{i,j})}{\sum_{i,k} \exp(l_{i,k})}$$

The weighted mutual information is then computed using these weights in place of the uniform weights used in standard mutual information calculations.

### C.2 Calibration Error

We compute the calibration error as the absolute difference between the mean confidence and mean accuracy. This is calculated by first computing the softmax of the logits to get probabilities, then taking the maximum probability as the confidence for each prediction. We then compare the predictions to the true labels to determine accuracy. The calibration error is the absolute difference between the mean confidence and mean accuracy across all samples.

### C.3 Metric Computation by Uncertainty Score

To analyze how different metrics vary with uncertainty, we compute metrics for different quantiles of an uncertainty score. This process involves:

1. Sorting the inputs based on the provided uncertainty scores.
2. Dividing the sorted inputs into quantiles.
3. Computing the specified metric for each quantile ("Bucket Average") or up to the given quantile ("Acceptance Threshold").

This approach allows us to observe how metrics like accuracy, negative log-likelihood, or calibration error change as a function of the model's uncertainty.

### C.4 Other Evaluation Metrics

In addition to the above, we used several standard evaluation metrics:

1. **Accuracy**: The proportion of correct predictions.
2. **Negative Log-Likelihood (NLL)**: The negative log-likelihood of the true labels under the model's predictions.
3. **Entropy**: The entropy of the model's predictive distribution, used as a baseline uncertainty measure for single models.
4. **Mutual Information**: For ensembles, we use the mutual information between the predicted class and the ensemble index as a measure of epistemic uncertainty.
5. **AUROC**: The Area Under the Receiver Operating Characteristic curve, used for evaluating out-of-distribution detection performance.

These metrics were computed across different uncertainty quantiles to analyze how model performance and uncertainty estimates correlate. For the AUROC calculations, we used the uncertainty scores (entropy for single models, (weighted) mutual information for ensembles) as the ranking criterion to distinguish between in-distribution and out-of-distribution samples.

## D Comparison with Fellaji & Pennerath (2024)

While we initially observed and documented the epistemic uncertainty collapse phenomenon in 2021, Fellaji & Pennerath (2024) independently discovered similar effects in Bayesian neural networks, terming it the "epistemic uncertainty hole". Our study offers several key extensions and insights:

- **Theoretical Framework:** We provide a theoretical explanation for the epistemic uncertainty collapse through the lens of ensembles of ensembles and propose the hypothesis of implicit ensembling for wider models, offering a mechanistic understanding of why this phenomenon occurs:
    - **Ensembles of Ensembles:** Our work introduces the concept of ensembles of ensembles, showing how the epistemic uncertainty collapse manifests in hierarchical ensemble structures. This provides a novel toy setting for the phenomenon not explored in the other work.
    - **Implicit Ensemble Extraction:** We propose and evaluate a novel technique for mitigating the epistemic uncertainty collapse through implicit ensemble extraction. This practical approach to addressing the issue goes beyond the observational nature of Fellaji & Pennerath (2024)'s work.

- **Broader Model Architectures:** While Fellaji & Pennerath (2024) primarily focus on MLPs, we demonstrate that this phenomenon extends to more complex architectures, including state-of-the-art vision models based on ResNets and Vision Transformers.

Thus, while Fellaji & Pennerath (2024) observe an epistemic uncertainty collapse, our work provides a more comprehensive theoretical and empirical investigation of this phenomenon. We not only confirm their findings across a broader range of models and datasets but also offer new insights into the mechanisms behind this effect and potential strategies for mitigation. The proposed implicit ensemble extraction represents an initial step towards addressing the practical challenges posed by the epistemic uncertainty collapse in real-world applications.

## E  Additional Results

Table 2: **Mean Mutual Information, Accuracy and NLL for Different MLP Widths and Datasets.**

| Dataset | Mean MI | | | Accuracy | | NLL | |
| --- | --- | --- | --- | --- | --- | --- | --- |
| MLP Width | MNIST | Dirty-MNIST | Fashion-MNIST | MNIST | Dirty-MNIST | MNIST | Dirty-MNIST |
| ×**1** | 0.0397 | 0.183 | 0.228 | 98 | 76.1 | 0.00216 | 0.477 |
| ×**2** | 0.0305 | 0.176 | 0.225 | 98.4 | 77.5 | 4.81e-05 | 0.531 |
| ×**4** | 0.0233 | 0.149 | 0.192 | 98.6 | 77.7 | 1.53e-06 | 0.471 |
| ×**8** | 0.0182 | 0.125 | 0.162 | 98.5 | 77.5 | 1.73e-06 | 0.415 |
| ×**32** | 0.011 | 0.0838 | 0.11 | 98.7 | 77.2 | 2.64e-07 | 0.402 |
| ×**64** | 0.0104 | 0.0882 | 0.114 | 98.6 | 77.1 | 3.73e-08 | 0.46 |
| ×**128** | 0.00956 | 0.0805 | 0.11 | 98.5 | 76.6 | 7.08e-08 | 0.351 |

Table 3: **Covariance between the mutual information as uncertainty metric and performance metrics for different ResNet models and extracted ensemble sizes.** Higher absolute values indicate stronger relationships. The arrows denote which (neg) covariance is to be preferred.

| Model | Uncertainty Metric | Performance Metric Ensemble Size | Neg. Bucket Covariance Calibration Error ↓ | Neg. Acceptance Covariance Accuracy ↑ | Neg. Log-Likelihood ↓ |
|---|---|---|---|---|---|
| **Resnet152-V2** | **Entropy** | **Original** | -0.030 | 0.046 | -0.144 |
| | **Mutual Information** | **2×2** | -0.016 | 0.050 | -0.179 |
| | | **3×3** | -0.030 | 0.044 | -0.150 |
| | | **4×4** | -0.033 | 0.053 | -0.189 |
| | | **5×5** | -0.035 | 0.054 | -0.192 |
| | | **6×6** | **-0.037** | 0.053 | -0.185 |
| | | **7×7** | -0.031 | **0.054** | **-0.203** |
| **Wide_Resnet101_2-V2** | **Entropy** | **Original** | -0.030 | 0.044 | -0.141 |
| | **Mutual Information** | **2×2** | -0.019 | 0.049 | -0.174 |
| | | **3×3** | -0.031 | 0.039 | -0.139 |
| | | **4×4** | -0.038 | 0.051 | -0.187 |
| | | **5×5** | -0.038 | 0.052 | -0.188 |
| | | **6×6** | **-0.039** | **0.052** | -0.191 |
| | | **7×7** | -0.036 | 0.050 | **-0.194** |
| **Resnext101_64X4D** | **Entropy** | **Original** | -0.018 | 0.046 | -0.147 |
| | **Mutual Information** | **2×2** | -0.011 | **0.049** | **-0.162** |
| | | **3×3** | -0.019 | 0.042 | -0.145 |
| | | **4×4** | -0.022 | 0.042 | -0.156 |
| | | **5×5** | -0.021 | 0.043 | -0.156 |
| | | **6×6** | **-0.024** | 0.044 | -0.159 |
| | | **7×7** | -0.021 | 0.043 | -0.156 |

Table 4: **Covariance between the weighted mutual information as uncertainty metric and performance metrics for different ResNet models and extracted ensemble sizes.** Higher absolute values indicate stronger relationships. The arrows denote which (neg) covariance is to be preferred.

| | | | Neg. Bucket Covariance | Neg. Acceptance Covariance | |
| | | Performance Metric | Calibration Error ↓ | Accuracy ↑ | Neg. Log-Likelihood ↓ |
| Model | Uncertainty Metric | Ensemble Size | | | |
|---|---|---|---|---|---|
| **Resnet152-V2** | **Entropy** | **Original** | -0.030 | 0.046 | -0.144 |
| | **Weighted MI** | **2×2** | -0.032 | 0.036 | -0.134 |
| | | **3×3** | -0.039 | 0.038 | -0.151 |
| | | **4×4** | -0.037 | 0.038 | -0.144 |
| | | **5×5** | **-0.042** | **0.047** | **-0.178** |
| | | **6×6** | -0.038 | 0.044 | -0.159 |
| | | **7×7** | -0.035 | 0.020 | -0.062 |
| **Wide_Resnet101_2-V2** | **Entropy** | **Original** | -0.030 | 0.044 | -0.141 |
| | **Weighted MI** | **2×2** | -0.039 | 0.040 | -0.138 |
| | | **3×3** | -0.039 | 0.039 | -0.148 |
| | | **4×4** | -0.044 | 0.043 | -0.159 |
| | | **5×5** | **-0.047** | 0.042 | -0.163 |
| | | **6×6** | -0.043 | **0.049** | **-0.180** |
| | | **7×7** | -0.037 | 0.016 | -0.073 |
| **Resnext101_64X4D** | **Entropy** | **Original** | -0.018 | 0.046 | -0.147 |
| | **Weighted MI** | **2×2** | **-0.043** | 0.036 | -0.131 |
| | | **3×3** | -0.042 | 0.045 | -0.160 |
| | | **4×4** | -0.039 | 0.043 | -0.150 |
| | | **5×5** | -0.042 | 0.045 | -0.157 |
| | | **6×6** | -0.040 | **0.049** | **-0.171** |
| | | **7×7** | -0.023 | 0.009 | -0.034 |

Table 5: **Covariance between the mutual information as uncertainty metric and performance metrics for the MaxVit model and extracted ensemble sizes.** Higher absolute values indicate stronger relationships. The arrows denote which (neg) covariance is to be preferred.

| | | | Neg. Bucket Covariance | Neg. Acceptance Covariance | |
| | | Performance Metric | Calibration Error ↓ | Accuracy ↑ | Neg. Log-Likelihood ↓ |
| Model | Uncertainty Metric | Ensemble Size | | | |
|---|---|---|---|---|---|
| **Timm-Maxvit** | **Entropy** | **Original** | **-0.124** | **0.043** | **-0.228** |
| | **Mutual Information** | **10×10** | -0.098 | 0.032 | -0.187 |
| | | **11×11** | -0.097 | 0.031 | -0.183 |
| | | **12×12** | -0.101 | 0.033 | -0.197 |
| | | **13×13** | -0.102 | 0.034 | -0.199 |
| | | **14×14** | -0.104 | 0.035 | -0.203 |
| | | **15×15** | -0.105 | 0.036 | -0.207 |
| | | **16×16** | -0.103 | 0.037 | -0.211 |
| | | **2×2** | -0.083 | 0.040 | -0.217 |
| | | **3×3** | -0.103 | 0.034 | -0.200 |
| | | **4×4** | -0.084 | 0.020 | -0.130 |
| | | **5×5** | -0.100 | 0.029 | -0.176 |
| | | **6×6** | -0.094 | 0.025 | -0.155 |
| | | **7×7** | -0.100 | 0.031 | -0.184 |
| | | **8×8** | -0.096 | 0.033 | -0.193 |
| | | **9×9** | -0.091 | 0.028 | -0.161 |

Table 6: *Covariance between the weighted mutual information as uncertainty metric and performance metrics for the MaxVit model and extracted ensemble sizes.* Higher absolute values indicate stronger relationships. The arrows denote which (neg) covariance is to be preferred.

| Model | Uncertainty Metric | Performance Metric
Ensemble Size | Neg. Bucket Covariance
Calibration Error ↓ | Neg. Acceptance Covariance
Accuracy ↑ | Neg. Log-Likelihood ↓ |
|---|---|---|---|---|---|
| **Timm-Maxvit** | **Entropy** | **Original** | **-0.124** | **0.043** | -0.228 |
| | **Weighted MI** | **10×10** | -0.089 | 0.041 | -0.236 |
| | | **11×11** | -0.089 | 0.041 | -0.235 |
| | | **12×12** | -0.086 | 0.041 | -0.236 |
| | | **13×13** | -0.088 | 0.041 | -0.237 |
| | | **14×14** | -0.086 | 0.041 | -0.236 |
| | | **15×15** | -0.085 | 0.041 | -0.235 |
| | | **16×16** | -0.082 | 0.043 | -0.233 |
| | | **2×2** | -0.093 | 0.039 | -0.213 |
| | | **3×3** | -0.115 | 0.042 | -0.237 |
| | | **4×4** | -0.101 | 0.041 | -0.236 |
| | | **5×5** | -0.100 | 0.041 | -0.236 |
| | | **6×6** | -0.095 | 0.040 | -0.234 |
| | | **7×7** | -0.094 | 0.040 | -0.234 |
| | | **8×8** | -0.085 | 0.041 | **-0.238** |
| | | **9×9** | -0.091 | 0.041 | -0.237 |

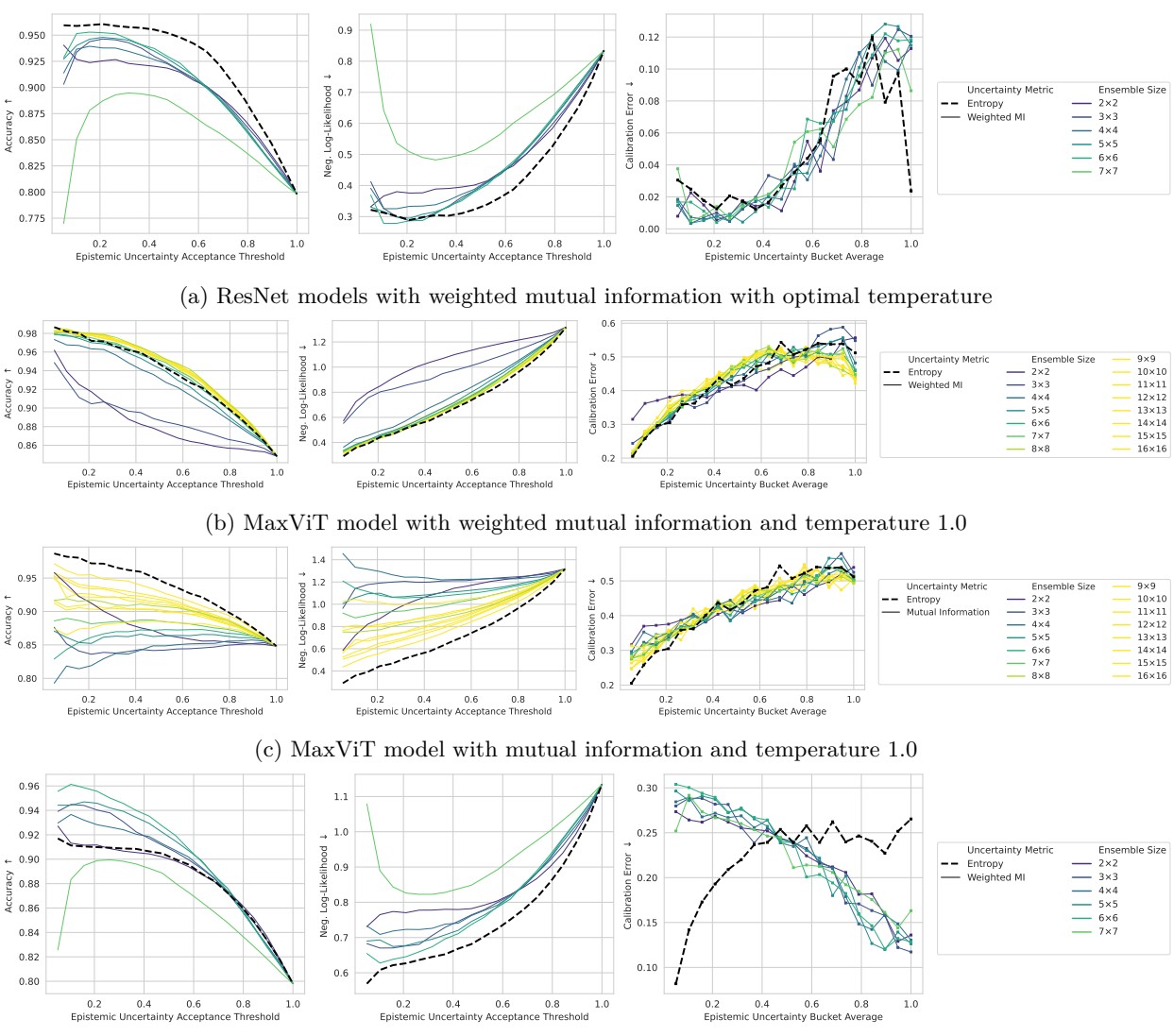

(a) ResNet models with weighted mutual information with optimal temperature

(b) MaxViT model with weighted mutual information and temperature 1.0

(c) MaxViT model with mutual information and temperature 1.0

(d) ResNet models with weighted mutual information and temperature 1.0

Figure 7: **Complementary Plots of Performance metrics for different ensemble sizes extracted from pre-trained models.**

