# OpenReview forum: "(Implicit) Ensembles of Ensembles: Epistemic Uncertainty Collapse in Large Models"
_TMLR — Accepted by TMLR_

### Review · Reviewer_VnwL · 2025-01-02

**Summary Of Contributions:**

The authors observe the collapse of epistemic uncertainty when using deep ensemble in the regression and classification task. They argue that large neural network models can suffer from implicit ensemble problem and may have a poor estimation of epistemic uncertainty.

**Audience:**

Yes

**Claims And Evidence:**

No

**Requested Changes:**

1. Provide clear evidence that the "epistemic uncertainty collapse" and "implicit emsemble" problem are not consequences of neural collapse observed by [1].
2. Provide a more detailed discussion on different mathematical evaluations of epistemic uncertainty.
3. Clarify the meaning of $p(\theta_m|D)$ used in Eq. 5 and further justify the approximation with mathematical evidences.

P.S. From my experience, the "epistemic uncertainty collapse" or "implicit ensembling" is very likely to be the consequence of the "neural collapse". Therefore, please provide clear evidence to differentiate the two phenomena.

**Strengths And Weaknesses:**

Pros:
1. The paper is well-written.
2. The figures look good.

Cons:

1. The author(s) should avoid using “We initially observed this behavior in 2021” in footnote 1, as the followed link could potentially give the reviewer the information of the authors and thus violate the double-blind review requirement.
2. The observation of epistemic uncertainty collapse and the implicit ensembling of the neural network models is very likely to be the consequence of neural collapse observed by [1] and explored widely in the follow-up works ([1] has 500+ citations reported by Google Scholar). The author(s) should provide clear evidence that these two phenomena are different to justify the contribution of the paper.
3. The mathematical definition of epistemic uncertainty is not unique, the paper used the definition in [3] while [2] has a different definition. I would recommend that the author(s) provide a more detailed discussion of the existing mathematical definitions of epistemic uncertainty to avoid confusion.
4. The approximation in Eq. 5 is not convincing. Firstly, [4] does not provide formal mathematical proofs to justify the approximation in Eq. 5. Secondly, in the deep ensemble approach, the ensemble members are trained individually over the train set with different initializations. Therefore the $p(\theta_m|D)$ used after Eq. 4 has a very different meaning from the Monte-Carlo dropout approach or sampling $\theta$ from the Bayesian prior. What is the practical meaning of $p(\theta_m|D)$ in the ensemble approach? Why is valid to use $p(\theta_m|D)$ when computing the mutual information? These questions require detailed clarification to justify the theoretical contribution of the paper.
5. Eq.9 holds if the approximation in Eq. 5 is correct and Eq. 10,11,12,13 looks a bit trivial.  It would be better if this theoretical evidence could be further explored.

Reference:

[1] “Prevalence of neural collapse during the terminal phase of deep learning training” Papyan et al., 2020

[2] “What Uncertainties Do We Need in Bayesian Deep Learning for Computer Vision? ” Alex Kendall and Yarin Gal, 2017

[3] “Understanding Measures of Uncertainty for Adversarial Example Detection” Lewis Smith and Yarin Ga, 2018

[4] “The power of ensembles for active learning in image classification” Beluch et al., 2018

---

> ### Author Response · Authors · 2025-02-13
> **Thank you for your review**
>
> Dear Reviewer VnwL,
>
> Thank you for your thoughtful comments. We address each point below:
>
> 1. Regarding the anonymized URL: We have checked and the link points to `ANON.notion.site`, which is anonymized and not the right domain and cannot reveal the authors' identity. This is not immediately evident without following the link, and we thank the reviewer for bringing this up.
>
> 2. On the relationship to neural collapse: While neural collapse (Papyan et al., 2020) describes the geometry of final-layer features in classification tasks (classes aligning with simplex vertices and features forming tight class-dependent clusters), our observed phenomenon is fundamentally different. Our epistemic uncertainty collapse also appears in regression tasks and non-neural models like random forests (see Figure 1 and the added Figure 6 in §A.4 in the appendix). Furthermore, neural collapse requires near-zero training loss in classification, while uncertainty collapse manifests across various scenarios, including partial training and data-sparse regression. We have clarified these distinctions and also shortly compare to feature collapse (van Amersfoort et al., 2020) in the related work section.
>
> 3. On epistemic uncertainty definitions: We have expanded our discussion to compare to the variance-based definition from Kendall & Gal (2017) in §A.1 in the appendix. Our use of mutual information builds on established work in active learning and Bayesian experimental design. While we explain why this metric is natural for our analysis, we acknowledge alternative approaches with additional citations.
>
> 4. Regarding Equation (5): We have revised the paper to clarify that our framework does not require posterior approximation. This should help side-step concerns. The epistemic uncertainty definition, equivalent to the expected information gain, does not need posterior samples but can be applied to any distribution. We now define p(θ) simply as the distribution of models obtained through empirical training with different random seeds (and thus model initialization and data shuffles). We include the posterior approximation perspective in the appendix with appropriate citations for completeness.
>
> 5. On Equations (9)-(13): Based on the overall feedback, we have restructured the proof to be more intuitive in the main text while providing a more detailed and more formal argument in the appendix in §A.3. We hope this new presentation better serves readers at different levels of technical depth.
>
> We appreciate your careful review and believe these changes have strengthened the paper.

---

### Review · Reviewer_2swD · 2025-01-22

**Summary Of Contributions:**

This paper challenges the common convention that larger neural networks are better at estimating epistemic uncertainty. The author's core assumption is that large models are simply "implicit ensemble of ensembles". Based on this key assumption, the authors demonstrate theoretically and empirically that larger ensembles of ensembles are worse at estimating epistemic uncertainty.

**Audience:**

Yes

**Claims And Evidence:**

No

**Requested Changes:**

As discussed in the weakness section, either show:
1. Toy experiment comparing epistemic uncertainty using a growing number of $N$ ensembles that take the same input which would be more consistent with implicit ensembles in large models.
2. Theoretically demonstrate how hidden transformations of the input work with Equations 7-13, instead of only raw inputs.

Minor changes:
1. The font of $\mathcal{M}$ is inconsistent in places, for example in Equation 6 ($M$) and 7 ($\mathcal{M}$).

**Strengths And Weaknesses:**

**Strengths**

**S1**: The paper investigates an important and under-explored topic of epistemic uncertainty in very large neural networks. This brings attention to an important issue with practical implications.

**S2**: The paper is clearly written and well organized, containing numerous helpful figures and diagrams. It introduces many concepts well which makes it accessible to a broad audience.

**S3**: Experiments use a variety of datasets, model architectures to verify their claims and provides an analysis to explain the observed phenomenon.

**Weakness**

My main concern with this paper is the loose connection between "large models" and "ensembles of ensembles", as argued on page 5.
While the paper's interpretation of successive layers in large models as "ensembles of ensembles" is interesting, it doesn't fit the formal definition of an ensemble. Traditionally, each member of the ensemble processes the same input. In contrast, successive layers in a neural network operate on the transformed outputs of previous layers, not a shared raw input. This definition is also shared by the author as per **Equation 7** in Section 3.

This subtle difference weakens the theoretical connection between the paper's explicit ensembles of ensembles experiments and the hypothesized implicit ensembling within large neural networks.
Specifically, the paper partitions an ensemble of $K\times M$ models into $K$ ensembles of $M$ models and observes a decrease in epistemic uncertainty.
This experimental setup doesn't align with the implicit ensembling within a single large model because each member layer does not share the same input. To convincingly support their claim, the authors should ideally show uncertainty collapse across a growing pool of $N$ models that take the same input, which would be more consistent with the implicit ensembles in large models.

Another way to address this would be to explain why hidden transformations of the input are consistent with Equations 7-13, which currently assumes the same raw input to every model.

---

> ### Author Response · Authors · 2025-02-13
> **Thank you!**
>
> Thank you for your review and the essential points you raise. Let us address your key concerns:
>
> ## Regarding the interpretation of "ensembles of ensembles" in large models
>
> We appreciate your concern about the analogy and agree the explanation needed more precision. The earlier drafting of the section was unlucky. Let us clarify the precise relationships we wanted to convey:
>
> - EoE shows epistemic uncertainty (EU) collapse across an ensemble of models that themselves are ensembles
> - We demonstrate that an ensemble of wide DNNs also exhibits EU collapse as the width increases
> - Importantly, we have clarified section §3.4 not to claim that each layer represents a different ensemble of ensembles. Rather, each model can be viewed as an implicit ensemble due to its width, and when we create an ensemble of such wide models (an ensemble of implicit ensembles), we observe similar EU collapse patterns as in explicit EoE
>
> Our empirical evidence supports this view:
>
> - In §4, we observe that ensembles of MLPs suffer from EU collapse as the width increases
> - In §4.1, we see how MLPs can be decomposed into sub-networks that recover useful EU (Figure 4)
> - In vision models, removing global average pooling reveals per-tile predictions that function as natural sub-models also recovering useful EU (Figure 5)
>
> In these experiments, all models and extracted sub-models take the same inputs.
>
> ## Regarding notation consistency
>
> We have standardized the font usage throughout all equations now (hopefully).
>
> We believe these clarifications better articulate the relationship between explicit and implicit ensembling. Please let us know if any points need further clarification.

---

### Review · Reviewer_GeaP · 2025-02-04

**Summary Of Contributions:**

The paper investigates epistemic uncertainty collapse, a phenomenon where larger deep learning models underestimate uncertainty due to implicit ensemble within their architectures. The authors provide a “theoretical explanation” for this effect, which I believe is incorrect. Through extensive experiments on MLPs, ResNets, and Vision Transformers, they empirically confirm this collapse and propose Implicit Ensemble Extraction, a method that decomposes large models into diverse sub-models to recover epistemic uncertainty. Their findings highlight the limitations of uncertainty estimation in large models and offer a method to restoring uncertainty estimates without retraining, with implications for safer and more reliable AI systems.

**Audience:**

Yes

**Claims And Evidence:**

No

**Requested Changes:**

To my understanding, the observed epistemic uncertainty collapse may be attributed to the following reasons: 1) The ensembles obtained from multiple training processes might not be truly i.i.d., leading to biased plug-in estimators that fail to converge properly. 2)The plug-in estimator itself may introduce substantial bias, which could be mitigated through improved estimation techniques. In addition, there is a long-standing body of literature on mutual information estimation, though the paper does not sufficiently engage with these established methods. It might be an interesting path to explore.

Some other recent literature on understanding the issue of epistemic uncertainty

Bengs, Viktor, Eyke Hüllermeier, and Willem Waegeman. "Pitfalls of epistemic uncertainty quantification through loss minimisation." Advances in Neural Information Processing Systems 35 (2022): 29205-29216.

Bengs, Viktor, Eyke Hüllermeier, and Willem Waegeman. "On second-order scoring rules for epistemic uncertainty quantification." In International Conference on Machine Learning, pp. 2078-2091. PMLR, 2023.

Shen, Maohao, Jongha Jon Ryu, Soumya Ghosh, Yuheng Bu, Prasanna Sattigeri, Subhro Das, and Gregory W. Wornell. "Are Uncertainty Quantification Capabilities of Evidential Deep Learning a Mirage?." In The Thirty-eighth Annual Conference on Neural Information Processing Systems. 2024.

**Strengths And Weaknesses:**

Strength:
The Empirical results are adequate to convince the significance of the issue of current ensembles based methods.

Weakness:

1. The derivation from (11) to (12) is incorrect. Since entropy is a concave function of its input, the order of expectation and entropy cannot be swapped in the second term. As the number of ensembles approaches infinity, the estimate will converge to the true mutual information, provided that each ensemble is independently and identically distributed (i.i.d.) sampled. This convergence follows from the Law of Large Numbers rather than the Central Limit Theorem.

2. The left-hand side of equation (8) is not well-defined, or at least, it is unclear why the two quantities are equal. It seems that the authors are attempting to use a plug-in estimator to approximate the quantity on the right-hand side, but this requires further clarification.

3. As a result, regardless of the empirical observations, the claimed explanation for epistemic uncertainty collapse does not appear to be valid. The phenomenon may stem from other underlying factors that have not been accounted for in the paper.

---

> ### Author Response · Authors · 2025-02-13
> **Thank you!**
>
> We thank reviewer GeaP for their helpful comments. We address their concerns below:
>
> 1) "The derivation from (11) to (12) is incorrect because entropy is concave, so we cannot simply swap the limit and the entropy."
>
> Our argument relies on the continuity of the entropy functional, not Jensen's inequality. Both the Monte Carlo average of the predictive distribution and the corresponding entropy converge to their expectations. We have revised the main paper to present a more intuitive argument based on the independence of random variables and the mutual information being zero then. We have added an extended proof that expands on the old argument with additional clarifications in §A.3.
>
> 1) "The left-hand side of Equation (8) seems not well-defined."
>
> We have updated the exposition: We use the distribution $p(\theta)$ over trained model parameters obtained through multiple independent training procedures (random initializations, data shuffling). Each trained model within an EoE represents a sample from this distribution. We have removed equation (8), updated the main paper to focus on an empirical ensemble sample distribution, and simplified the argument. The appendix then considers the mutual information between predictions and parameter distributions.
>
> 1) "The ensembles might not be truly i.i.d. or the plug-in estimator may be biased."
>
> Indeed, ensembles from random initializations aren't guaranteed to be perfect posterior samples. We have removed mentions of data conditioning from the main paper and focus on the distribution of models after training with different random seeds. We examine prior work on deep ensembles and Bayesian approximation (Mandt et al., 2017; Wilson & Izmailov, 2020) in §A.2. We have updated the main paper to focus on empirical ensemble sample distribution without requiring the ensembles to be perfect posterior samples. The appendix further considers the mutual information between predictions and parameter distributions.
>
> 1) Literature suggestions regarding epistemic uncertainty quantification.
>
> We thank the reviewer for suggesting Bengs et al. (2022, 2023) and Shen et al. (2024). We've incorporated these citations in the theory section, and we will update the related work section further for the camera-ready version.
>
> We believe this clarifies the concerns raised by the reviewer. Please let us know if you have any other feedback.

---

> ### Comment · Reviewer_GeaP · 2025-03-03
>
> I appreciate the authors' revisions—this version is much clearer than the initial draft.
>
> The key takeaway from the proposed theoretical framework is reflected in Equation (11). Specifically, the first MI term captures the disagreement across sub-ensembles, which vanishes as we adopt more complex models according to the author. Meanwhile, all epistemic uncertainty is captured by the second MI term, which represents the disagreement within each sub-ensemble. In practice, if we estimate epistemic uncertainty solely by training large models multiple times, we only account for the first term, which ultimately vanishes.
>
> To my understanding, the core issue is that the learned model $\theta$ cannot be interpreted as a sample from the posterior distribution $P(\theta | D)$. While the proposed algorithm further divides each model into submodels, it does not convincingly establish that this approach provides a better approximation to the posterior compared to the original ensemble approach.
>
> A sound theoretical framework should demonstrate the following:
>
> 1) That increasing model complexity indeed induces an implicit ensemble effect, leading to reduced variation in the predictive distribution.
>
> 2) That the submodel approach effectively addresses this issue by generating better posterior approximation, and it provides clear guidelines on how to separate the models.
>
> The current theoretical framework only justifies the claim that ensembles of ensembles lead to collapsed epistemic uncertainty (which is relatively obvious), but it does not establish a strong connection to why it happens when we are training larger models.
>
> More broadly, I would argue that ensemble methods are not inherently effective for uncertainty quantification, as learning algorithms, e.g., SGD, are not designed for sampling from the posterior distribution. This paper could serve as an insightful blog post illustrating the limitations of ensembles in certain scenarios.

---

> > ### Author Response · Authors · 2025-03-03
> > **Response to Reviewer GeaP**
> >
> > Thank you for your additional feedback. We appreciate your acknowledgment that the revised version is clearer. We would like to address several points in your latest comment:
> >
> > 1. **On posterior distributions**: As mentioned, we have deliberately removed any requirement for models to be samples from a posterior distribution in our theoretical framework. Our analysis focuses solely on the empirical distribution of trained models with different random seeds. This approach sidesteps the concerns you raised about SGD not being designed for posterior sampling. We still have added a section §A.2 that details peer-reviewed literature on the connection between SGD and posterior sampling.
> >
> > 2. **On model complexity and implicit ensembling**: Our experimental results directly demonstrate the connection between increasing model complexity and epistemic uncertainty collapse:
> >
> >    * Figure 2 shows uncertainty collapse in explicit EoE
> >    * Figure 3 demonstrates the same effect occurs when increasing model width
> >    * Figure 4 empirically validates that we can decompose wide models into sub-models that recover epistemic uncertainty
> >    * Figure 5 extends this to vision models, where removing global average pooling reveals per-tile predictions that function as natural sub-models
> >
> >    These results establish a clear empirical link between model width/complexity and uncertainty collapse, supporting our theoretical framework.
> >
> > 3. **On submodel approach**: Section 4.1 provides detailed guidelines on how to separate models into submodels, with specific approaches for MLPs and vision models. Our empirical evidence demonstrates that this approach effectively recovers some epistemic uncertainty that would otherwise be lost.
> >
> > 4. **On ensembles for uncertainty quantification**: While we agree there are potential theoretical limitations to ensemble methods, despite all the published peer-reviewed literature in §A.2, extensive empirical evidence from multiple studies shows that deep ensembles consistently provide state-of-the-art uncertainty quantification in practice (e.g., Ovadia et al., 2019; Beluch et al., 2019), often outperforming more computationally intensive Bayesian methods.
> >
> > The practical implications of our work extend well beyond a blog post. We offer actionable insights for improving uncertainty quantification in real-world machine learning systems where deep ensembles are widely deployed, and we identify the reason for Fellaji and Pennerath's epistemic uncertainty hole. Our method for extracting implicit ensembles from single models provides a practical approach to recovering epistemic uncertainty that would otherwise be lost as model complexity increases.

---

### Decision · Action_Editor_zNas · 2025-03-16

**Recommendation:** Accept with minor revision

**Comment:**

Please See the Original Decision and Commentary Below

UPDATE: I have conducted further consultation with two of the reviewers. The feedback from both was that the paper could be edited to an acceptable form if the authors made a MAJOR change to the claims in the introduction and abstract (and in a couple of sentences discussing results). While this is a major change to the claims, it would still constitute a minor change to the paper. I have therefore adjusted the recommendation to "Accept with minor revision".

The reviewers wrote: "this paper is very well written and has interesting results" and that they "would support the paper if the author modified their claims", and that "if 1) the argument in the manuscript on the connection between “epistemic uncertainty collapse” and “implicit emsembling” is revised to be not so strong ... I am fine for the acceptance of the paper." There were reiterated concerns that "Theoretical analysis on the problem is a bit trivial (and not rigorous)" but there was not an argument that the theoretical analysis was incorrect.

If the claims are modified carefully throughout the paper to the following, then the paper could be accepted: CR1: We introduce implicit ensembling as a possible explanation for the phenomenon of uncertainty collapse in wide/large models. We provide theoretical analysis and experiments that demonstrate uncertainty collapse in ensembles of ensembles and experimental evidence of collapse in wide/large models. CR2: We provide a method for extracting an ensemble from a model and show that this extraction procedure can recover the uncertainty.

The current following claims are problematic because the evidence and analysis in the paper does not support them sufficiently. They should be removed or rewritten: C1: “We provide a possible explanation for this phenomenon through our theoretical analysis and experiments demonstrating implicit ensembling, along with methods to recover epistemic uncertainty by extracting implicit ensembles from large models”. - while most of this claim is supported (it is a “possible” explanation), it is not clear that any experimental results do genuinely demonstrate implicit ensembling. C2: “we provide evidence for both epistemic uncertainty collapse and implicit ensembling” - There is no clear experimental evidence of implicit ensembling. C3: [Experiments]: “These results provide compelling evidence for the epistemic uncertainty collapse due to implicit ensembling in neural networks.” and “This shows that epistemic uncertainty collapse due to implicit ensembling might be a pervasive phenomenon in neural networks, even in relatively simple MLP models.” - The results do not provide clear evidence that the epistemic uncertainty collapse is due to implicit ensembling (or even that implicit ensembling is definitey occurring), so these conclusions are not supported.

In light of this additional feedback from the reviewers, I support acceptance provided that the authors modify their claims as outlined above (to CR1 and CR2, removing/editing C1-C3.)

I will carefully check the modified paper to ensure that the claims have been suitably adjusted.

_______________________________________
________________________________________
---------------------------------------------------
The reviewers appreciated the authors’ effort in explaining the results and revising the paper and there were comments that the paper was well-written. The reviewers acknowledged that the revisions had addressed initial concerns regarding the theoretical framework.

However, all three reviewers recommended rejection of the paper, expressing concerns that the connection between wide networks and “ensembles of ensembles” is insufficiently strong to justify the claims and conclusions of the paper.

The reviewers acknowledged that the experiments demonstrated epistemic uncertainty collapse. The consensus view is that the theoretical framework justifies the claim that ensembles of ensembles lead to collapsed epistemic uncertainty, but all the reviewers considered that this is fairly obvious and the derivation of this results is relatively straightforward.

All reviewers were concerned that the paper’s theoretical development and experimental results do not provide compelling evidence that the underlying reason for the uncertainty collapse phenomenon is the presence of ensembles of ensembles in larger networks. All the reviewers considered that the connection is not sufficiently established, and there would need to be additional theoretical analysis or carefully designed experiments to establish this.

Although my personal view of the paper is more positive, it is not reasonable to override the recommendations of three reviewers.

Section 3, the theoretical part of the paper, establishes that ensembles-of-ensembles lead to uncertainty collapse. Section 3.4 provides the hypothesis that “large neural networks could be viewed as implicit ensembles” and that “an ensemble of such large neural networks could thus form an ensemble of implicit ensembles”. There is no theoretical support for this hypothesis, but the introduction is written in such a way that it suggests that the paper does provide sufficient evidence for this hypothesis. We are thus left to consider whether the empirical results in Section 4 are sufficiently compelling to support the claim.

Section 4 commences by showing experimentally that ensembles of ensembles do lead to uncertainty collapse (Figures 1 and 2). The paper then explores the training of small MLP models of varying width on MNIST. The results demonstrates that as the width of the MLP increases, there is a clear trend of decreasing mutual information. The paper the claims that these results “provide compelling evidence for the epistemic uncertainty collapse due to implicit ensembling”. This is a key component of the paper where the conclusion appears to overreach the experimental evidence. The experiments do not explicitly investigate or establish implicit ensembling. The results show only that as the width increases, the mutual information decreases. This could be due to implicit ensembling but there could be many other explanations.

Section 4.1 provides results showing that it is possible to mitigate the collapse by decomposing the larger model into constituent submodels. Although these results do provide some support for the implicit ensembling hypothesis, the extraction process does involves steps such as “maximiz[ing] the diversity among the resulting sub-models by regularizing the mutual information”, suggesting that the connection between the hypothesized implicit ensemble in the original model and the extracted ensemble is not so simple.
Indeed, the paper can only conclude that the results “provide support for our hypothesis” rather than “establish our hypothesis”. This contrasts with the stronger statement in the introduction that “we offer both theoretical justification and empirical evidence for the epistemic uncertainty collapse phenomenon”. Perhaps the reviewers’ concerns with the paper can be addressed by toning down the introductory claims. I would strongly encourage the authors to revise and resubmit the work, since the hypothesis and supporting results are very interesting.

**Audience:**

Some individuals would be interested in knowing the findings. The paper addresses an important topic and proposes a very interesting hypothesis.

**Claims And Evidence:**

The paper claims to "provide evidence for both epistemic uncertainty collapse and implicit ensembling" and "offer both theoretical justification and empirical evidence for the epistemic uncertainty collapse phenomenon".

All reviewers are of the opinion that the paper falls short of establishing these claims. The reviewers acknowledged that the experiments demonstrated epistemic uncertainty collapse. The consensus view is that the theoretical framework justifies the claim that ensembles of ensembles lead to collapsed epistemic uncertainty.

However, all reviewers expressed concerns that the paper does not provide sufficient evidence to support its hypothesis that reason for the uncertainty collapse phenomenon is the presence of ensembles of ensembles in larger networks.

---

> ### Author Response · Authors · 2025-05-06
> **Thank you**
>
> Dear Action Editor and Reviewers,
>
> Thank you very much for your thorough review of our paper "Ensemble of Ensembles: Epistemic Uncertainty Collapse in Large Models." We are grateful for your constructive feedback and the recommendation for acceptance with minor revisions.
>
> We appreciate the clear guidance on modifying our claims to better align with the evidence presented in the paper. We will revise the manuscript to position implicit ensembling as a possible explanation for uncertainty collapse rather than making stronger claims about definitive evidence.
>
> Thank you again for your time and careful consideration of our work.
>
> Best regards

---

> > ### Author Response · Authors · 2025-05-11
> > **PTAL**
> >
> > Dear Action Editor and Reviewers,
> >
> > We have uploaded a revision with the proposed changes. We have also added a subsection on a speculative connection to Neural Tangent Kernel theory. A PDF difference between the two revisions is here: https://draftable.com/compare/OJKnCXmNZrZM
> >
> > Some examples of the implemented changes (*not* exhaustive):
> >
> > ---
> >
> > > "While these patterns are consistent with the implicit ensembling hypothesis we have proposed, we note that other explanations might also account for these observations."
> >
> > > "These results demonstrate epistemic uncertainty collapse in wider neural networks. This pattern is consistent with our implicit ensembling hypothesis, though other explanations may also be possible."
> >
> > > "While this is consistent with the implicit ensembling hypothesis, it does not prove that implicit ensembling is the mechanism responsible for uncertainty collapse in wide networks."
> >
> > > "Future work should investigate the exact mechanisms behind epistemic uncertainty collapse in large models [...]"
> >
> > ---
> >
> > Please let us know if the changes are as requested. We will then upload the camera-ready revision.
> >
> > Again, our thanks to you and the reviewers so much for these suggestions.
> >
> > Best wishes

---

> > > ### Comment · Action_Editor_zNas · 2025-05-19
> > > **Revision acceptable - please upload the camera-ready revision**
> > >
> > > The modifications have sufficiently changed the claims such that the majority of the reviewers would be satisfied that the paper establishes the presented claims, and I am also of this opinion.
> > >
> > > Please submit the camera-ready version.